# Structural insights into ligand recognition and activation of the medium-chain fatty acid-sensing receptor GPR84

Heng Liu[1,9], Qing Zhang [1,2,3,4,9], Xinheng He [1,5,9], Mengting Jiang[6], Siwei Wang[1,4,5], Xiaoci Yan[1,2,5], Xi Cheng [1,5], Yang Liu[1,4], Fa-Jun Nan[1,2,3,4], H. Eric Xu [1,5,6,7] ✉, Xin Xie [1,2,3,4,5] ✉ & Wanchao Yin [1,5,8] ✉

GPR84 is an orphan class A G protein-coupled receptor (GPCR) that is predominantly expressed in immune cells and plays important roles in inflammation, fibrosis, and metabolism. Here, we present cryo-electron microscopy (cryo-EM) structures of $G\alpha_i$ protein-coupled human GPR84 bound to a synthetic lipid-mimetic ligand, LY237, or a putative endogenous ligand, a medium-chain fatty acid (MCFA) 3-hydroxy lauric acid (3-OH-C12). Analysis of these two ligand-bound structures reveals a unique hydrophobic nonane tail -contacting patch, which forms a blocking wall to select MCFA-like agonists with the correct length. We also identify the structural features in GPR84 that coordinate the polar ends of LY237 and 3-OH-C12, including the interactions with the positively charged side chain of R172 and the downward movement of the extracellular loop 2 (ECL2). Together with molecular dynamics simulations and functional data, our structures reveal that ECL2 not only contributes to direct ligand binding, but also plays a pivotal role in ligand entry from the extracellular milieu. These insights into the structure and function of GPR84 could improve our understanding of ligand recognition, receptor activation, and $G\alpha_i$-coupling of GPR84. Our structures could also facilitate rational drug discovery against inflammation and metabolic disorders targeting GPR84.

GPR84, still considered an orphan G protein-coupled receptor (GPCR), has been reported to be activated by medium-chain fatty acids (MCFAs) and a range of synthetic compounds[1,2]. GPR84 is mainly expressed in immune cells[3], including monocytes, macrophages, and neutrophils in the periphery and microglia in the brain, and has long been identified as a proinflammatory receptor. Accordingly, GPR84 is considered a potential therapeutic target in a range of diseases, including ulcerative colitis, fibrotic diseases, nonalcoholic steatohepatitis, and acute respiratory distress syndrome[4–8]. Indeed, GPR84 was reported to be upregulated by certain inflammatory stimuli, such as lipopolysaccharide (LPS) and $TNF\alpha$[9–11], and GPR84 activation in macrophages leads to increased cytokine secretion, chemotaxis,

[1]State Key Laboratory of Drug Research, Shanghai Institute of Materia Medica, Chinese Academy of Sciences, Shanghai, China. [2]School of Pharmaceutical Science and Technology, Hangzhou Institute for Advanced Study, University of Chinese Academy of Sciences, 310024 Hangzhou, China. [3]Shandong Laboratory of Yantai Drug Discovery, Bohai Rim Advanced Research Institute for Drug Discovery, 264117 Yantai, Shandong, China. [4]National Center for Drug Screening, Shanghai Institute of Materia Medica, Chinese Academy of Sciences, 201203 Shanghai, China. [5]University of Chinese Academy of Sciences, 100049 Beijing, China. [6]School of Chinese Materia Medica, Nanjing University of Chinese Medicine, 210023 Nanjing, China. [7]School of Life Science and Technology, ShanghaiTech University, 201210 Shanghai, China. [8]Zhongshan Institute for Drug Discovery, Shanghai Institute of Materia Medica, Chinese Academy of Sciences, 528400 Guangdong, China. [9]These authors contributed equally: Heng Liu, Qing Zhang, Xinheng He. ✉e-mail: eric.xu@simm.ac.cn; xxie@simm.ac.cn; wcyin@simm.ac.cn

and phagocytosis[12]. Considering the pathophysiological role of GPR84 signaling, several antagonists have been developed, including GLPG1205, PBI-4050, and PBI-4547[1], which have displayed therapeutic effects in animal models of inflammatory and fibrotic diseases and are being evaluated in clinical studies. In addition, GPR84 plays a crucial role in eye development in *Xenopus laevis*[12] and regulates mitochondrial function and quality control in murine skeletal muscle[13]. GPR84 also possesses macrophage-mediated antiatherosclerosis properties similar to those of GPR109A, and agonists of GPR84, including embelin and its derivatives, have been shown to display antiatherogenic effects[14]. These findings indicate that GPR84 plays complex roles under physiological and pathological conditions and that GPR84 ligands with activating or inhibitory effects may both have potential clinical applications.

Free fatty acids (FFAs), comprising hydrocarbon chains terminating with carboxylic acid groups, are not only essential fuel sources for humans and animals but also important signaling molecules that contribute to many cellular functions. MCFAs, particularly those with 10-12 carbon acids (decanoic acid (C10), undecanoic acid (C11), and lauric acid (C12) and their 2-hydroxy or 3-hydroxy forms, such as 3-OH-C12[15], are suggested to be endogenous ligands of GPR84[2,16]. Due to the modest potency of MCFAs and their low levels circulating in the plasma, it is still difficult to view them as bona fide physiological ligands of GPR84. In efforts to better understand the physiological role of GPR84, several synthetic agonists have been discovered[2]. 6-n-Octylaminouracil (6-OAU) was the first synthetic GPR84 agonist discovered by screening[15]. It was more potent than MCFAs, including undecanoic acid and lauric acid, in inducing cytoskeletal or adhesion changes in lipopolysaccharide (LPS)-stimulated primary murine macrophages[15,17]. We also identified a GPR84 agonist (2-(hexylthio)-pyrimidine-4,6-diol, also called ZQ-16) by high-throughput screening[18]. A subsequent SAR study led to the identification of 6-nonylpyridine-2,4-diol (Cpd 51, also called LY237) as a potent and selective GPR84 agonist with an $EC_{50}$ of 0.189 nM[19], which is the most potent GPR84 agonist reported to date[1].

Here, we present cryo-EM structures of $G\alpha_i$ protein-coupled human GPR84 in complex with LY237 or the putative endogenous ligand, 3-OH-C12, respectively. The structures reveal that ECL2 with a downward-movement conformation, contributes to the ligand binding and plays a pivotal role in ligand entry from the solvent. Additionally, our structures present features of the orthosteric ligand-binding pocket for MCFA recognition, and activation mechanisms of GPR84.

## Results

### The overall structure of the LY237-GPR84-$G\alpha_i$ complex

To understand the ligand binding and receptor activation of GPR84, we first focused on the most potent synthetic agonist LY237. To obtain a stable LY237-bound $G\alpha_i$-coupled GPR84 complex (GPR84-$G\alpha_i$), we introduced a BRIL tag at the N-terminus (NT) of the full-length wild-type (WT) GPR84 and applied the NanoBiT tethering strategy to improve complex stability and homogeneity[20]. The GPR84-$G\alpha_i$ complex was further stabilized by scFv16[21], which was introduced to bind to the interface between $G\alpha_i$ and $G\beta$ (Supplementary Fig. 1). The structure was finally determined to an overall resolution of 3.2 Å using single-particle cryo-EM (Fig. 1a and Supplementary Fig. 1). Most EM density maps of the complex were well resolved, enabling unambiguous modeling of the complex, including the ligand LY237, the $G\alpha_i$ protein heterotrimer, scFv16, and the majority of amino acid side chains of the receptor GPR84, while BRIL was absent from the map because of high flexibility, which is a common result in other GPCR structure determinations (Supplementary Fig. 2).

The overall conformation of the GPR84-$G\alpha_i$ complex is similar to most known structures of lipid receptors (Fig. 1b). The extracellular loops (ECLs) of GPR84, mainly ECL2, together with the N-terminus, form a roof-like structure, hanging over the LY237 binding pocket,

rendering the orthosteric site inaccessible from the extracellular milieu (Fig. 1c, d). It should be pointed out that the ECL1/3 and the NT exhibit an organized style of stretched loops, whereas the ECL2 presents a canonical β-sheet conformation, which shifts downwards compared to that in other lipid receptors, such as FFA1, S1PR1, LPA1, and CB2[22–25] (Fig. 1c, d, and Supplementary Fig. 3a). LY237 within the binding pocket stabilizes the active conformation of GPR84, which forms a large intracellular transmembrane domain (TMD)cavity to couple the heterotrimeric $G\alpha_i$ protein complex. The detailed interactions will be described below. Unless otherwise specified, all the mechanisms discussed throughout our paper refer to the LY237-GPR84-$G\alpha_i$ complex.

### The unique disulfide bridge combination in GPR84

It is well known that disulfide bonds in the extracellular domain confer structural stability of the receptors to coordinate ligand access and binding in most GPCRs, especially for lipid receptors[26]. Further structural analysis revealed two disulfide bonds in the extracellular domain of the GPR84 structure (Fig. 1c). One is the highly conserved disulfide bond[25] between C93[3.25] and C168[ECL2], which makes the ECL2 in GPR84 shift downwards and pack directly against LY237, and stabilizes ECL2 for LY237 binding. This TM3-ECL2 disulfide bond is also observed in most lipid GPCRs of known structures, including EP2-4, LPA6, FFA1, PAFR, CysLT1-2, and BLT1[26,27]. Another unexpected disulfide bond is formed between cysteine-11 from the N-terminus (C11[NT]) and cysteine-166 from ECL2 (C166[ECL2]) (Fig. 1c, d). This non-conserved disulfide bond is unusual in lipid receptors, but was also predicted by Alphafold in the previous studies[28,29], which suggest its potential role for GPR84 activation. Indeed, this NT-ECL2 disulfide bond in GPR84 pulls the ECL2 toward TM1, which completely occludes the orthosteric site in GPR84 from the extracellular milieu. Destruction of this NT-ECL2 disulfide bond by alanine substitutions of either single mutants or double mutations in our mutational assays completely abolished the activation of GPR84 (Fig. 1e), indicating the critical role of the disulfide bond between ECL2 and NT in GPR84 activation. To our knowledge, the only other lipid receptor that has an NT-ECL2 disulfide bond is LPA1 (Supplementary Fig. 3b), which does not contain the conserved TM3-ECL2 disulfide bond[30]. ECL2 of LPA1 adopts a loop conformation but only partially covers the ligand binding pocket from the extracellular side, while the NT in LPA1 folds into a short helix and forms the rest of the roof, assisted by the NT-ECL2 disulfide bond in LPA1, to completely cover the entry of ligand pocket. In summary, the NT-ECL2 disulfide bond contributes to the completely occluded extracellular pocket in both GPR84 and LPA1.

Considering the downward movement of ECL2 relative to other lipid receptors and the unique disulfide bridge combination in GPR84, we speculate that the TM3-ECL2 and NT-ECL2 disulfide bonds could govern the extracellular domain packing between ECL2 and the N-terminus and contribute to ligand binding and receptor activation. Overall, these features of the unique disulfide bridge combination in the extracellular domain of GPR84 provide fresh insights into the features of lipid receptors.

### LY237 binding and the amphipathic orthosteric pocket of GPR84

LY237 is an amphipathic molecule that has a hydrophilic head and a highly hydrophobic alkyl tail, mimicking the potential endogenous MCFA ligand. The orthosteric pocket of GPR84 is amphipathic to accommodate LY237 and can be divided into a polar region at the top (composed of an N-terminal cap and ECL2) and a deep hydrophobic cavity below (composed of TM3 to TM7) (Fig. 2a). This pattern of the ligand-binding pocket was also conserved in lipid receptors, including LPA receptors and S1PR receptors[25,29–31]. The detailed interactions are described below (Fig. 2b-e). In the upper polar sub-pocket, the polar head of LY237 is deeply buried in a polar pocket capped by ECL2 rather

than exposed to the extracellular environment, as mentioned above. The two hydroxyl groups form extensive polar interactions with S169[ECL2], R172[ECL2], and W360[7.43] (Ballesteros–Weinstein numbering in superscript[32]) (Fig. 2b, c). In addition, there are aromatic-dependent interactions to accommodate the binding of the LY237 pyridine ring, including the pi-pi interaction between the LY237 pyridine ring and F335[6.51] (Fig. 2d), the cation-pi interaction between R172[ECL2] and F170[ECL2], and the aromatic packing between Y69[2.53] and W360[7.43] (Fig. 2c). The van der Waals interactions formed by the LY237 pyridine ring with F101[3.33] and V165[ECL2] were also observed (Fig. 2d). Consistent with these structural observations, mutations at the residues described above

reduced LY237-induced GPR84 activation. Furthermore, both alanine substitutions of R172[ECL2] and F335[6.51] nearly abolished GPR84 activation, and mutations of other ligand-bound related residues mentioned above affect the activation of GPR84 to different extents (Fig. 2f, g and Supplementary Table 1), indicating their pivotal roles in GPR84 activation.

In the deep hydrophobic cavity, the hydrophobic tail of LY237 is wrapped by the surrounding amino acids, including L100[3.32], F101[3.33], I108[3.40], F152[4.57], L182[5.42] and F335[6.51] (Fig. 2b) (the Ballesteros-Weinstein numbering for helix 5 will be discussed below), with the long carbon chain projecting into the cleft between TM3 and TM5, which is sealed

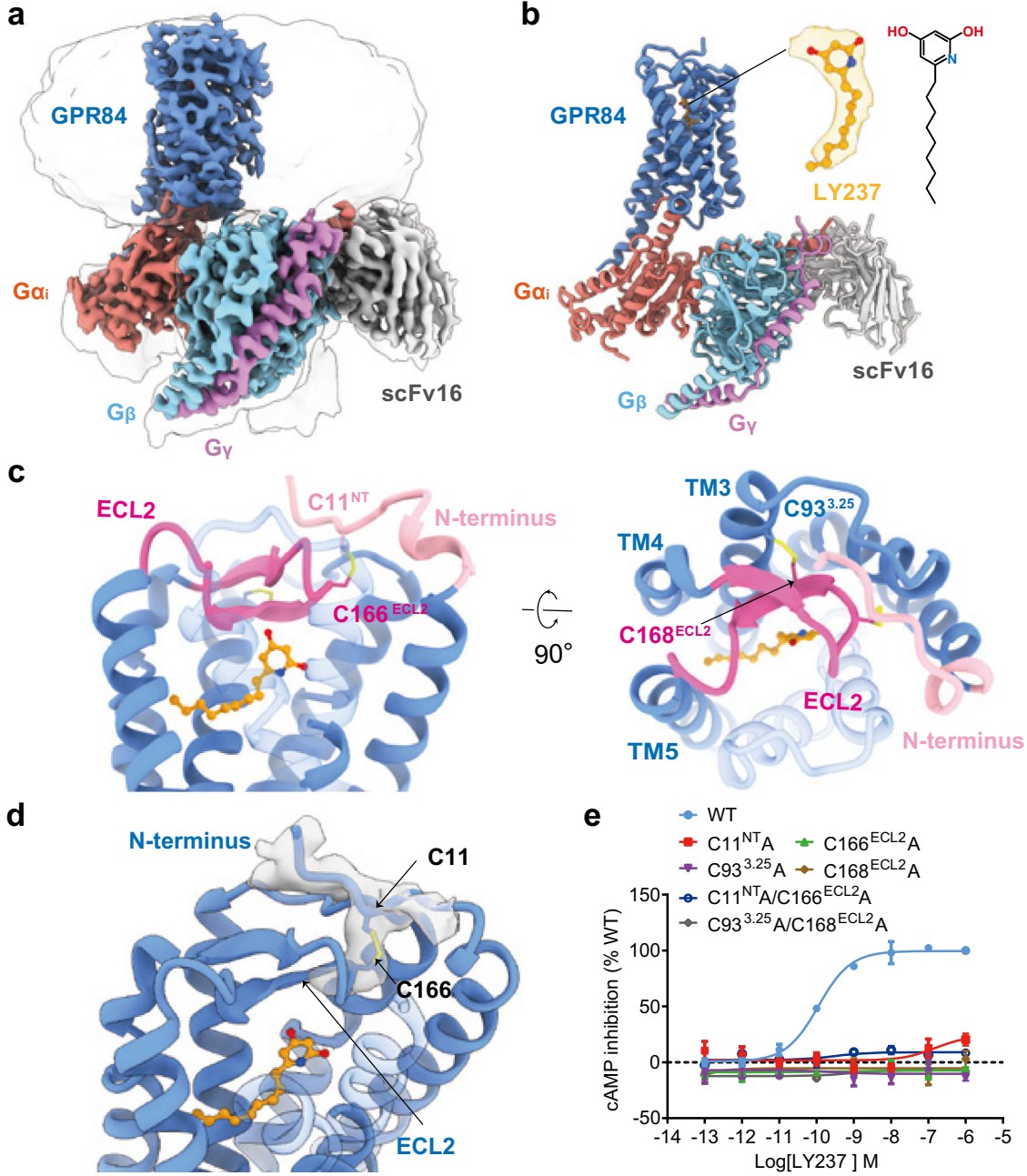

**Fig. 1 | Overall structure and unique features of LY237-GPR84-Gα_i complex. a, b** Cryo-EM density map (a) and ribbon presentation (b) of the LY237-GPR84-Gα_i complex. The complex density map is shown at a level of 0.06, and the micelle density map is shown at a level of 0.15. **c** Side view (left panel), and top view (right panel) of the distinct ECL2 (hot pink), N-terminus (light pink) conformation and disulfide bridge of GPR84. **d** The local density and ribbon presentation of the C11[NT]-

C166[ECL2] disulfide bridge. **e** Dose-response curves of LY237 in activating GPR84 with mutations in the disulfide bridges with cAMP assay. Data are shown as mean ± S.E.M. from a minimum of three technical replicates, which performed in triplicates. The representative dose-response curves are shown. Source data are provided as a Source data file.

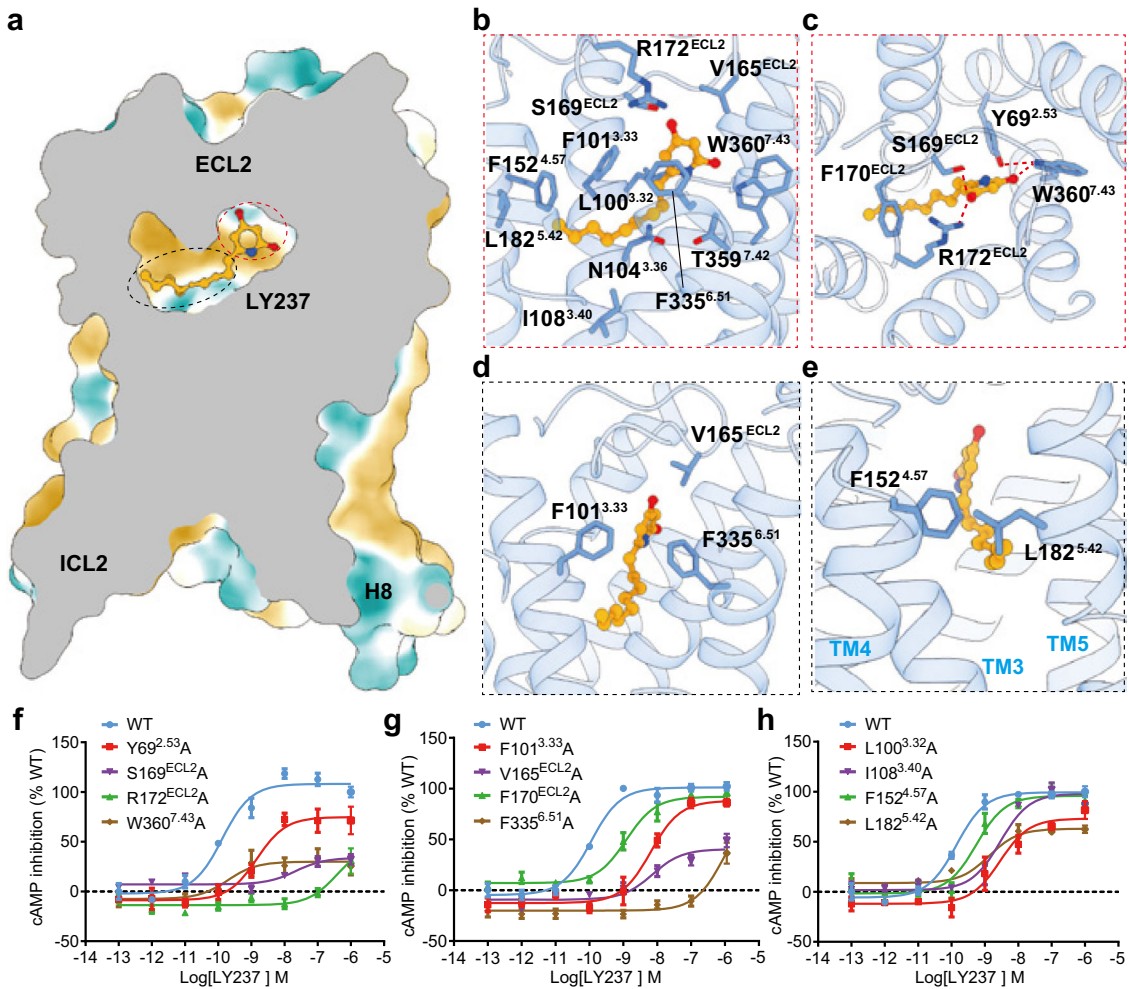

**Fig. 2 | LY237-binding pocket of GPR84. a** Hydrophilic (blue) and hydrophobic (yellow) properties of LY237-binding pocket of GPR84. **b** Detail interactions between LY237 and GPR84. **c** Highlighted polar interactions between the polar head of LY237 and GPR84. **d** Non-polar interactions between the pyridine head of LY237 and key residues from GPR84. **e** Hydrophobic tail of GPR84 sat at the cleft between TM3-5 and hold by F152[4.57], L182[5.42]. **f–h** Dose-response curves of LY237 in activating GPR84 with mutations in the binding pocket with cAMP assay. Data are shown as mean ± S.E.M. from a minimum of three technical replicates, which performed in triplicates. The representative dose-response curves are shown. Source data are provided as a Source data file.

by F152[4.57], and L182[5.42] (Fig. 2d, e). This interaction pattern is consistent with our mutational assays that alanine substitutions of these residues moderately reduced receptor activation by approximately 4-50-fold (Fig. 2g, h and Supplementary Table 1). Together with the limitation of the upper hydrophilic pocket against the polar end of the LY237, the wall formed by F152[4.57] and L182[5.42] helps to define the length of the binding pocket, which may explain why the potency of a GRP84 ligand is highly sensitive to the length of the alkyl tail[1,2].

**Structural identification of 3-OH-C12 as a GPR84 ligand**
As mentioned above, the orthosteric site in LY237-bound GPR84 is completely occluded from the extracellular milieu by the downward movement of ECL2 relative to other lipid receptors. To determine conformational changes of GPR84 induced by LY237 binding, we attempted to solve the structure of GPR84 in the apo state. We assembled and purified the GPR84-Gα$_i$ complex with no ligand supplied, and determined the structure to a resolution of 3.24 Å (Fig. 3a and Supplementary Fig. 1). The overall structure of this complex is very similar to that of the LY237-GPR84 complex.

Surprisingly, a narrow but extended shape of EM density occupied the orthosteric ligand-binding pocket, with a slight difference in orientation from LY237 (Supplementary Fig. 2). The observed density map was fitted well by MCFAs with carbon lengths of C10-C12 or their

metabolites with hydroxy substitutions (Supplementary Fig. 2c). To determine the exact identity of the bound ligand, we first performed mass spectrum (MS) analyses on the GPR84-Gα$_i$ complex with no ligand supplied used for our structural studies. However, only fatty acids with carbon length of more than 14 carbon atoms are identified, without any short- or medium-chain fatty acids with 3-OH substituents (Supplementary Data 1). Then, we tested several fatty acids with acyl tail lengths from C8 to C20 using cAMP assays to determine the effects of free fatty acids on GPR84 activation. Our results showed that both C10 and C12 could activate GPR84 well, especially the 3-OH-C12 had the highest potency with an EC$_{50}$ of 24.7 μM (Fig. 3b and Supplementary Table 1). Coincidentally, the EM density map of the compound found in the solved GPR84 structure with no ligand supplied matches well with the conformation of the 3-OH-C12 (Fig. 3a). Considering GPR84 could be activated by MCFAs or their hydroxylated forms[15], in order to better understand 3-OH-C12 recognition by GPR84, we successfully assembled the GPR84 complex activated by adding exogenous 3-OH-C12 at a concentration of 200 μM (about 8-fold higher than the EC$_{50}$ of 3-OH-C12 on GPR84) throughout the complex preparation process. Finally, we obtained the 3-OH-C12 bound GPR84 complex at an overall resolution of 2.89 Å using single-particle cryo-EM (Supplementary Fig. 4). The 3-OH-C12 in this structure matches well with our previous model solved without ligand applied (Fig. 3a, c and

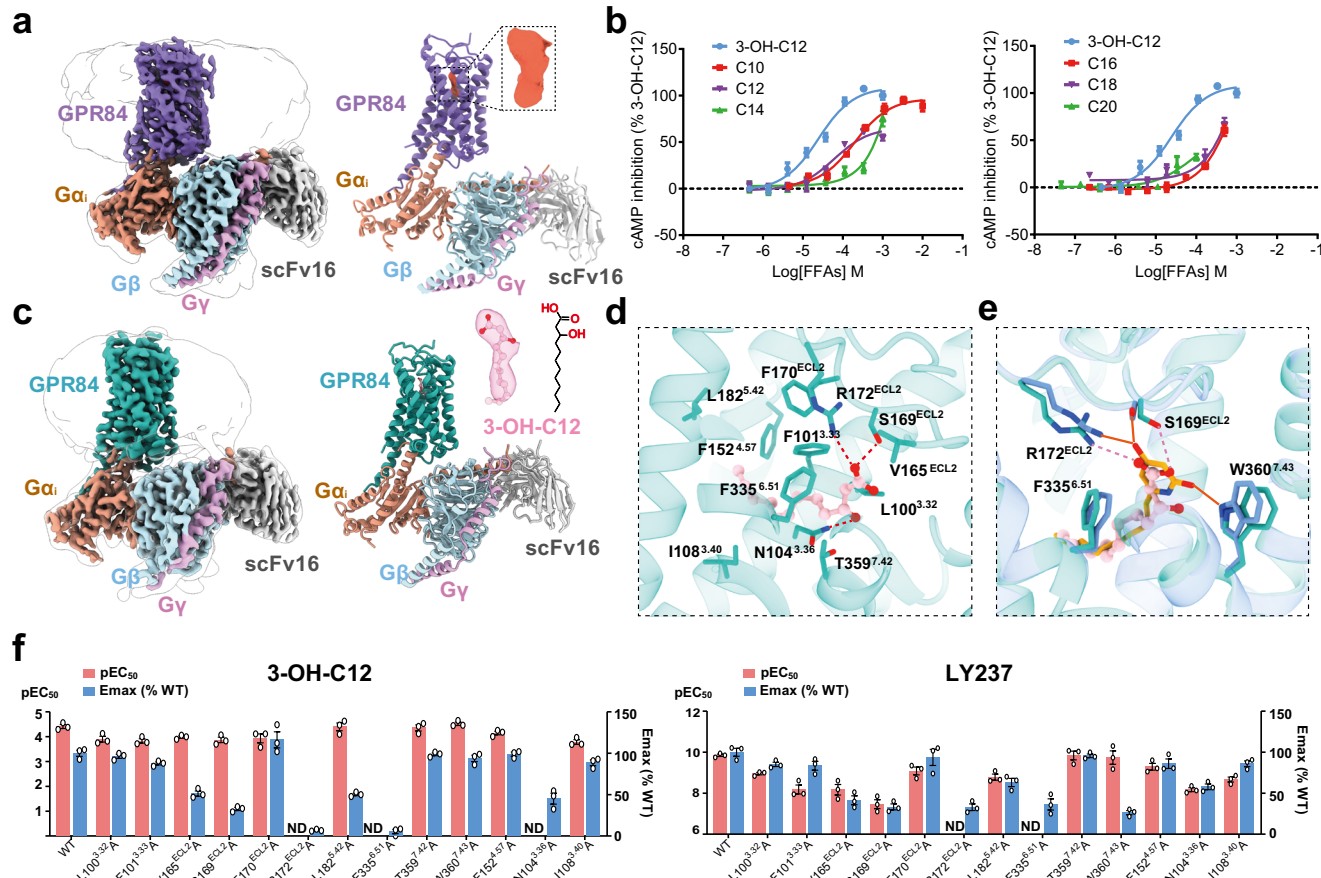

**Fig. 3 | Identification of the 3-OH-C12 bound GPR84-Gα$_i$ complex by cryo-EM. a** Cryo-EM density map (left) and ribbon presentation (right) of the GPR84-Gα$_i$ complex without any ligand added, the additional map in the binding pocket is shown on the upper right (tomato). **b** Dose-response curves of free fatty acids (FFAs) with different chain lengths in inducing GPR84-mediated cAMP inhibition, with 3-OH-C12 as control. Data are shown as mean ± S.E.M. from a minimum of three technical replicates, which performed in triplicates. The representative dose-response curves are shown. C10, decanoic acid; C12, lauric acid; C14, myristic acid; C16, palmitic acid; C18, stearic acid; C20, arachidic acid. **c** Cryo-EM density map (left) and ribbon presentation (right) of the 3-OH-C12-GPR84-Gα$_i$ complex.

**d** Detailed interactions of 3-OH-C12 with residues in the binding pocket of GPR84. **e** Interaction differences between LY237 (orange solid line) and 3-OH-C12 (pink dotted line) bound to GPR84. **f** Mutagenesis study to identify the key interaction residues for 3-OH-C12 (left panel) or LY237 (right panel) mediated GPR84 activation with cAMP assay. Values are shown as the mean ± S.E.M. from a minimum of three technical replicates, which performed in triplicates. ND-not determined, indicates that the activation level is too low to determine EC$_{50}$ values. Numerical data for graphs in **f** are available as in Supplementary Table 1. Source data are provided as a Source data file.

Supplementary Fig. 4). This 3-OH-C12 bound GPR84 complex presents better density for the bound 3-OH-C12 and the surrounding residues from GPR84 (Supplementary Fig. 4), which structurally provides a reliable template for MCFA recognition by GPR84, with detailed interactions with the receptor described below.

3-OH-C12 shares an identical nonane tail with LY237, and the nonane tail of 3-OH-C12 projects into the cleft between TM3 and TM5 in a conformation similar to LY237 (Fig. 3c). The hydrophobic interactions formed by the nonane tail and surrounding residues in GPR84 were validated by our mutational assays (Fig. 3d–f, Supplementary Fig. 5, and Supplementary Table 1). Meanwhile, the hydroxy substitution points to a hydrophilic pocket formed by Y69$^{2.53}$, N104$^{3.36}$, and T359$^{7.42}$ and forms a hydrogen bond with N104$^{3.36}$. These extensive polar interactions could promote the increased stability of interactions between 3-OH-C12 and GPR84, which may account for its higher potency compared with that of C12 (Fig. 3d). In our cAMP assays, 3-OH-C12 activated GPR84 with over 270,000-fold lower potency than LY237 (Supplementary Table 1). To determine the recognition mechanism of these two ligands by GPR84, we focused on the polar interactions in the up-polar subpocket. The carboxyl head of lauric acid forms a salt bridge with R172$^{ECL2}$ and a polar interaction with S169$^{ECL2}$, which can also be found in LY237 binding (Figs. 2b, c and 3d, e). In addition, the

negative head of lauric acid forms anion-π packing with F335$^{6.51}$ (Fig. 3d, e). Consistent with these structural observations, the S169$^{ECL2}$A mutation reduced 3-OH-C12-induced GPR84 activation by approximately fourfold, and replacing both R172$^{ECL2}$ and F335$^{6.51}$ with alanines almost abolished GPR84 activation (Fig. 3f and Supplementary Table 1).

Despite the presence of similar interaction pairs in both structures, the interactions with R172$^{ECL2}$ and S169$^{ECL2}$ were weaker in the 3-OH-C12-bound structure due to the longer contacts in the 3-OH-C12-bound structure than in the LY237-bound GPR84. In addition, analogous interactions between LY237 and W360$^{7.43}$ in the LY237 structure were not found in the 3-OH-C12-bound GPR84 (Fig. 3e), which may contribute to the tremendous difference in the activation potential of these two agonists for GPR84. Considering the pivotal roles of S169$^{ECL2}$, R172$^{ECL2}$, F335$^{6.51}$, and W360$^{7.43}$ in LY237-induced GPR84 activation (Figs. 2f and 3f), our structures reinforce the advantage of the polar end of aromatic rings with proper hydroxy substitutes and provide the basis for the potency differences of these two ligands on GPR84 activation.

**Potential ligand entry of GPR84 through ECL2 dynamics**
Both GPR84 structures have a completely sealed/occluded pocket. To investigate how ligand enters to the pocket, we performed ligand

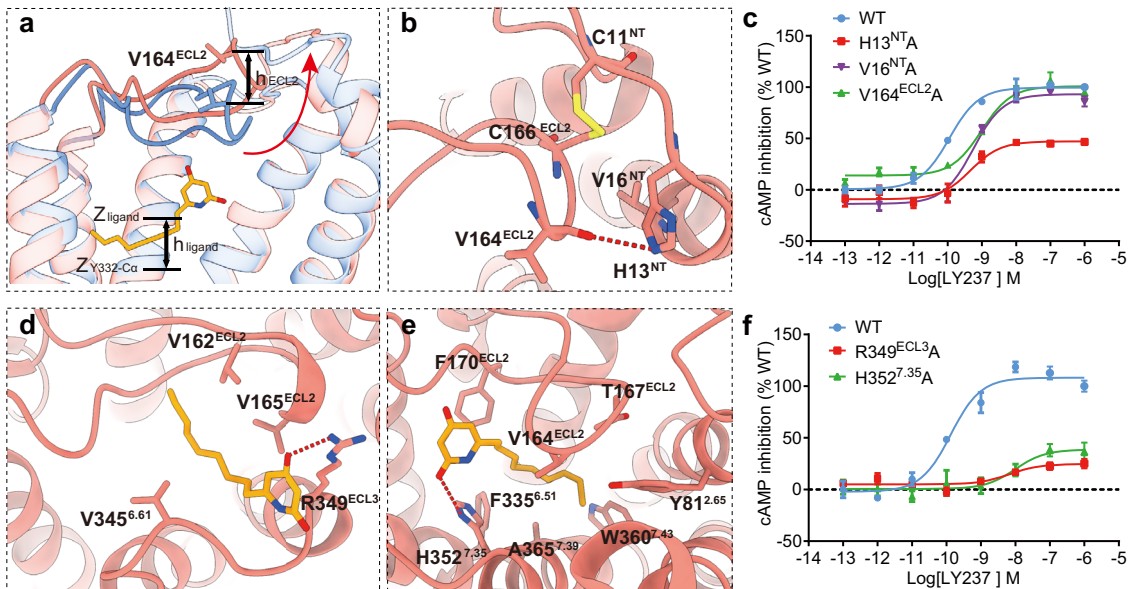

**Fig. 4 | MD simulation study of GPR84 ligand entry. a** Comparison of ECL2-opening conformation from molecular dynamics (salmon) and ECL2-closed conformation from EM data (cornflower blue) in GPR84. **b** Key interactions with ECL2 in the ECL2-opening conformation from molecular dynamics stimulation. **c** Dose-response curves of LY237 in activating GPR84 with point mutations affecting ECL2-opening interactions. Data are shown as mean ± S.E.M. from a minimum of three technical replicates, which performed in triplicates. The representative dose-response curves are shown. **d, e** Representative capture of the important interactions in intermediate state 1 (**d**) and intermediate state 2 (**e**) during LY237 entry. **f** Dose-response curves of LY237 in activating GPR84 with point mutations involved in interactions in IS1 and IS2 state. Data are shown as mean ± S.E.M. from a minimum of three technical replicates, which performed in triplicates. The representative dose-response curves are shown. Source data are provided as a Source data file.

Gaussian accelerated molecular dynamics (LiGaMD) simulations. LiGaMD adds a specific bias potential to the system, thus accelerates the ligand binding process[33]. With LiGaMD, we observed the ligand entry through the opened ECL2 (Fig. 4a, b, Supplementary Fig. 6, and Supplementary Movie 1). The difference in the Z coordinate between the Cα of $V164^{ECL2}$ in simulations and cryo-EM structure ($h_{ECL2}$) was used to measure the open of ECL2. Meanwhile, the difference in the Z coordinate between LY237 and the Cα of $Y332^{6.48}$ ($h_{ligand}$) were applied to depict its entry[34] (Supplementary Fig. 6a–c). The closed ECL2 in the cryo-EM structure spontaneously opened in 3 of 8 LiGaMD simulations, in which ECL2 shifted upwards by more than 2.5 Å, (Supplementary Fig. 6b). During ECL2 opening, the main chain of V164 forms a hydrogen bond with H13 (3.4 Å), and the disulfide bond between C166-C11 forms hydrophobic interactions with V16 (4.3 Å) (Fig. 4b). Alanine substitutions of H13, V16 and V164, to disrupt these interactions influenced the activation of GPR84, indicating that these interactions are important characteristics of open ECL2 trajectories (Fig. 4c).

Upon ECL2 opening, LY237 entered the binding pocket between ECL2 and TM7, through two representative intermediate states (IS1 and IS2) (Fig. 4d, e) in the free energy landscape from the unbound (U) state to the bonded (B) state (Supplementary Fig. 6a). From the U state to the IS1 state, $h_{ligand}$ changed from more than 22 Å to approximately 17-18.5 Å, indicating the process of ligand entry (Supplementary Fig. 6a). In the IS1 state, the polar head of LY237 formed hydrogen bonds with $R349^{ECL3}$ (2.8 Å), and its tail was attracted by the hydrophobic environment consisting of $V162^{4.67}$, $V165^{ECL2}$, and $V345^{6.61}$ (Fig. 4d). The importance of the potential LY237-$R349^{ECL3}$ interaction for GPR84 activation is underlined by the over 70% reduced $E_{max}$ and approximately 50-fold reduced potency of the ligand on the receptor carrying the R349A mutation (Fig. 4f). For the IS2 state, LY237 had an $h_{ligand}$ value between 11.2 Å and 15.2 Å (Supplementary Fig. 6a). The ligand in the IS2 state interacted with $H352^{7.35}$ in a hydrogen bond (3.5 Å), and its tail was trapped in a condensed hydrophobic environment made up of $Y81^{2.65}$, $V164^{ECL2}$, $T167^{ECL2}$, $F170^{ECL2}$, $F335^{6.51}$, $A365^{7.39}$, and $W360^{7.43}$ (Fig. 4e). This observation matched the speculation in a

previous study that the $R292^{7.35}$ in S1PR1 is a "cationic lure", projecting its side chain into the hydrophobic milieu of the membrane upper leaflet to attract phospholipids[26,35]. Consistent with the simulation observations, the $H352^{7.35}$A mutation reduced the potency of LY237 by approximately 40-fold and the $E_{max}$ by approximately 70% (Fig. 4f).

After the IS2 state, the ligand finally entered the pocket and adopted a pose similar to the cryo-EM pose (Supplementary Movie 1). The cryo-EM pose was also stable in the following conventional MD simulations, represented by the stable interaction between $R172^{ECL2}$ and LY237 and the root mean square deviation (RMSD) of 0.75-1.0 Å between MD and cryo-EM poses (Supplementary Fig. 6f, g). In summary, our simulations indicated the binding pathway of LY237 through the ECL2 open process, and the important interactions during ligand entry suggested by MD and mutagenesis studies may provide clues for understanding the dynamic process of lipid receptors.

## Activation mechanisms of GPR84

Comparison of our LY237-bound GPR84 structure with the previously reported antagonist-bound inactive S1PR1 structure[36] (PDB ID: 3V2Y) and the agonist-bound active structure of S1PR1[37] (PDB ID: 7EVY) sheds light on the mechanism of LY237-mediated GPR84 activation. The GPR84 structure adopts a fully active conformation, which is similar to that of active S1PR1 (Fig. 5a, b). Compared with inactive S1PR1, the cytoplasmic end of TM6 in GPR84 has a pronounced outward displacement, which opens the core of the receptor to allow $Gα_i$ engagement, along with an inward movement of the TM7 cytoplasmic end (Fig. 5a, b).

To study the allosteric transmission between the G protein-coupling cavity described above and the orthosteric ligand-binding pocket, we first set out to analyze the conserved motifs near the LY237-binding site of GPR84. It is well known that the ratchet-like movement of a triad of hydrophobic residues, termed PIF motif, (I3.40, P5.50, F6.44), has been observed in many active-state structures of family A GPCRs upon agonist-induced receptor activation[38–40]. Detailed structural analysis showed that almost all residues of speculative helix 5

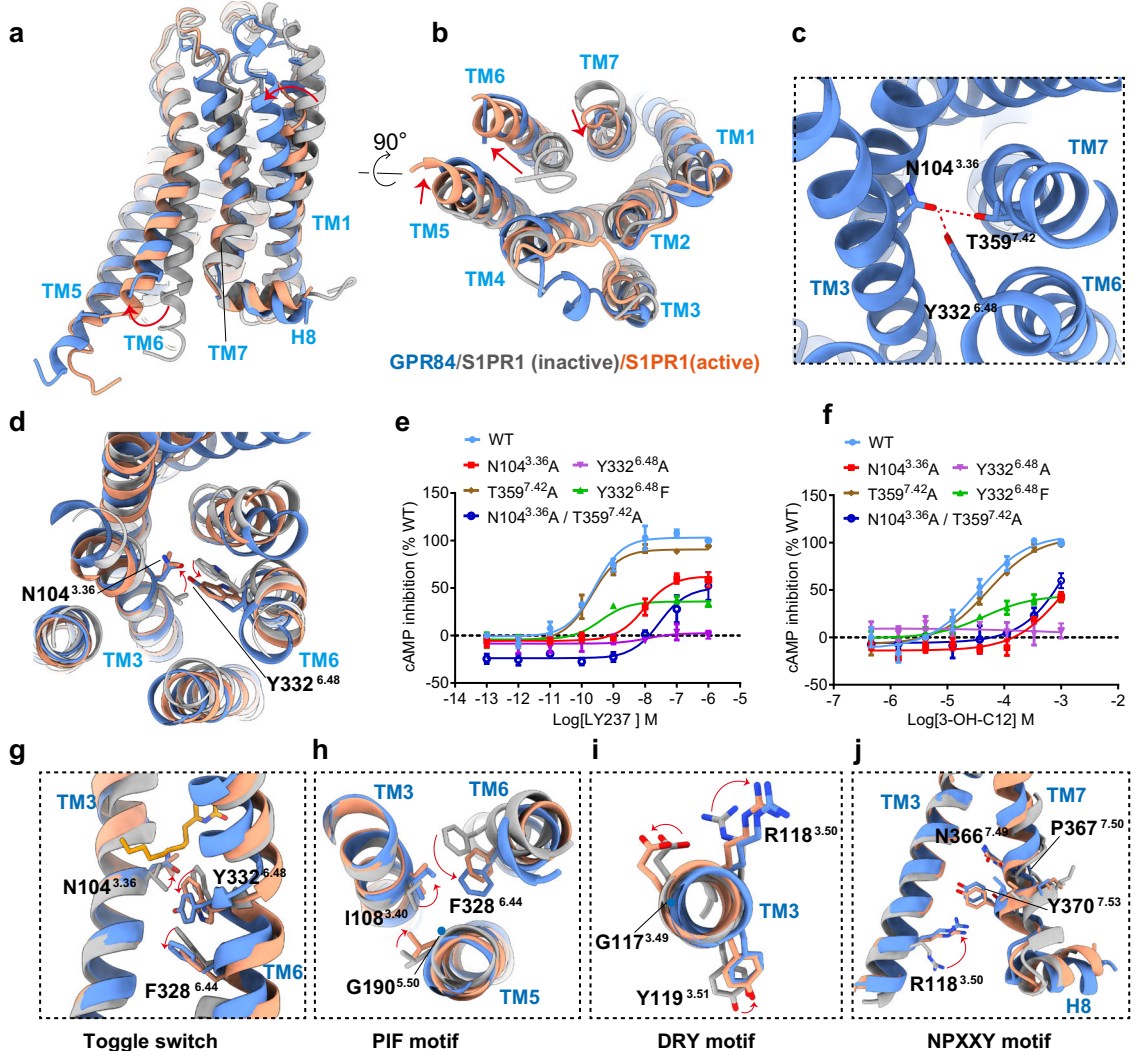

**Fig. 5 | GPR84 activation mechanism. a, b** Superposition of activated GPR84 (cornflower blue) with active S1PR1 (light salmon; PDB code: 7EVY) and inactive S1PR1 (gray; PDB code: 3V2Y). Notable conformational changes occur at intracellular ends of TM6 and TM7 upon receptor activation, side view (**a**) and top view (**b**). **c** N104³·³⁶ forms hydrogen bond with Y332⁶·⁴⁸ and T359⁷·⁴². **d** Detailed comparison of activated GPR84(cornflower blue) with inactive S1PR1 (gray; PDB code: 3V2Y) and active S1PR1 (light salmon; PDB code: 7EVY). In particular, side-chains N104³·³⁶ and Y332⁶·⁴⁸ exhibit similar displacement to L128³·³⁶, W269⁶·⁴⁸ of S1PR1 upon activation.

**e, f** Dose-response curves of LY237(e) and 3-OH-C12(f) in activating GPR84 with mutations destroying the hydrogen bonds. Data are shown as mean ± S.E.M. from a minimum of three technical replicates, which performed in triplicates. The representative dose-response curves are shown. **h–j** The "dual toggle switch" N104³·³⁶ and Y332⁴·⁴⁸ of GPR84 display relative rotameric change when sensing agonist. **h–j** The key G-I-F⁶·⁴⁴(PIF motif in common GPCRs) (**h**), G-R³·⁵⁰-Y(D(E)RY motif in common GPCRs) (**i**), and N-P⁷·⁵⁰-xx-Y⁷·⁵³ (**j**) motifs displayed conformational rearrangement in activated GPR84. Source data are provided as a Source data file.

(residues from D213 to V259, annotated in the GPCRDB database[41] (https://gpcrdb.org) were located in the region of ICL3, which is invisible in our solved structure (Supplementary Figs. 2 and 4). Based on structural alignment, the corresponding PIF motif is composed of G190 from TM5, I108 from TM3 and F328 from TM6 (Supplementary Fig. 7a). Consistently, helix 5 in GPR84 looks straight because of the G190⁵·⁵⁰ replacement, which is consistent with L⁵·⁵⁰/I⁵·⁵⁰ replacement in other lipid receptors[20,21] (Supplementary Fig. 7b). This is in contrast to a kink TM5 helix in other receptors possessing the conserved P⁵·⁵⁰, such as 5-HT₁AR and D2R[42,43].

It is noted that the toggle switch residue W⁶·⁴⁸, a highly conserved residue in class A GPCRs[39], is replaced by Y⁶·⁴⁸ in GPR84, which forms a hydrogen bond with the side chain of N104³·³⁶ (Fig. 5c). The conserved L(F)³·³⁶-W⁶·⁴⁸ residue pair in lipid receptors has been studied as a "dual toggle switch"[25,31,37], which has also been shown to play a key role in the activation of several other receptors, including AT1[44] and MC4[45]. Further structural analysis showed that the N104³·³⁶-Y332⁶·⁴⁸ residue pair in GPR84 adopts a similar conformation to L(F)³·³⁶-W⁶·⁴⁸ in the active S1PR

structure[25] (Fig. 5d), suggesting that GPR84 undergoes the common class A GPCR activation pathway, despite the divergence at position 6.48. In addition, N104³·³⁶ forms a hydrogen bond with the side chain of T359⁷·⁴² (Fig. 5c). Consistent with these structural observations, destruction of the hydrogen bond between N104³·³⁶ and Y332⁶·⁴⁸ in GPR84 by the single mutant N104³·³⁶A and double mutant N104³·³⁶A/T359⁷·⁴²A decreased GPR84 activation by approximately 50- and 200-fold and reduced the maximum activation to 50% of the wild-type receptor, respectively, despite T359⁷·⁴²A had minor impact on GPR84 activation by either LY237 or 3-OH-C12 (Fig. 5e, f). Moreover, Y332⁶·⁴⁸F, which retains the core phenyl group, reduced the maximum activation to less than 40% of the wild-type receptor, and Y332⁶·⁴⁸A totally abolished the activation (Fig. 5e, f). In summary, the hydrogen bond between N104³·³⁶ and Y332⁶·⁴⁸ plays an important role in receptor activation and makes N104³·³⁶-Y332⁶·⁴⁸ a unique "dual toggle switch" in GPR84.

The swing of N104³·³⁶-Y332⁶·⁴⁸ following ligand entry is accompanied by a marked displacement of F328⁶·⁴⁴ toward TM5 in the PIF

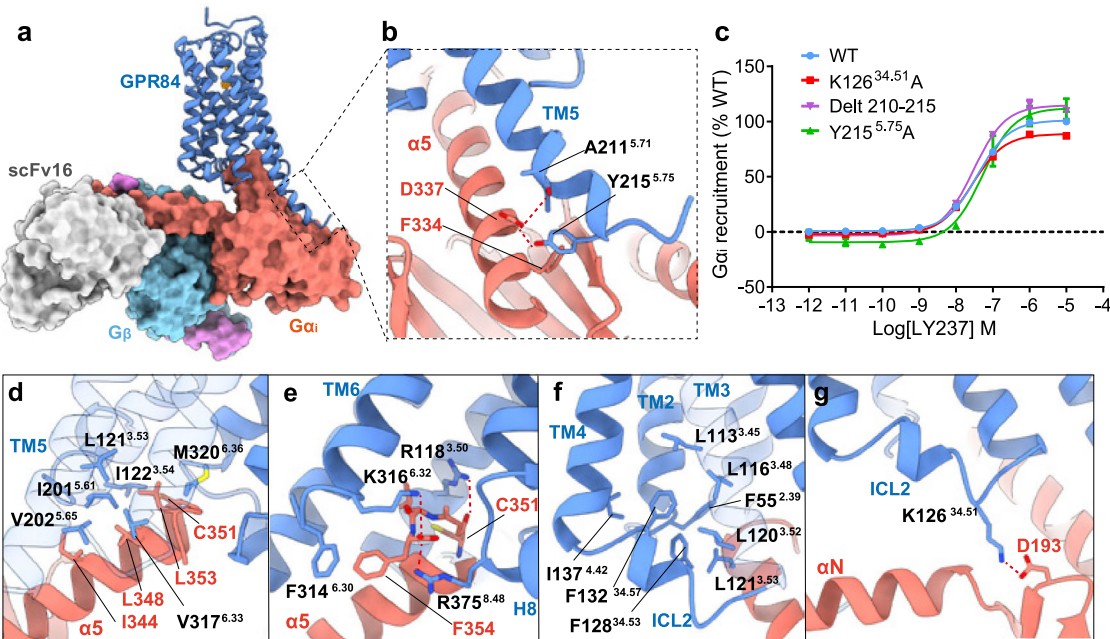

**Fig. 6 | Gαi coupling of GPR84. a** Cytoplasmic cavity formed by TM3, TM5, TM6, and TM7 of GPR84, with the C-terminal α5 helix of the Gαi inserted. **b** Key interactions between the extended TM5 and Gαi residues. **c** Dose-response curves of LY237 in inducing Gαi recruitment to GPR84 with various mutations. Data are shown as mean ± S.E.M. from a minimum of three technical replicates, which performed in triplicates. The representative dose-response curves are shown. **d, e** Hydrophobic network (**d**) and polar interactions (**e**) between GPR84 and α5 helix of Gαi. **f** Hydrophobic packing formed by residues from TM2-4 and ICL2. **g** Interactions formed by ICL2 and αN of Gαi protein. Source data are provided as a Source data file.

(GIF in GPR84) motif (Fig. 5g, h). The conformational changes are further associated with the displacement of the NPxxY motif toward the center of the core and upward of R118[3.50] in the conserved DRY (GRY in GPR84) motif in GPR84 (Fig. 5i, j). The new contacts of R118[3.50], Y198[5.58] and Y370[7.53] stabilized the agonist-bound conformation (Supplementary Fig. 8). Together, these conformational changes open up the TMD cavity to allow the α5 helix of Gαi to insert into the TMD core of GPR84.

**The identical G protein coupling mode of GPR84**

An inspection of the overall and detailed interactions between GPR84 and the Gαi heterotrimer under detergent micellular conditions revealed both conserved and unique coupling modes. First, the overall structure of the GPR84-Gαi complex was reminiscent of the interaction modes of the SP1R-Gαi and CB2-Gαi complexes[37,46]. The Gαi proteins in these receptor-Gαi complexes have a similar clockwise rotation (Supplementary Fig. 9a) when viewed from the intracellular to extracellular direction. The side chain of the D337 from Gαi forms two hydrogen bonds with the main chain carbonyl group of A211[5.71] and the side chain of Y215[5.75] in the extended TM5 of GPR84 (Fig. 6a, b). Our mutational analysis showed no obvious divergence in Gαi recruitment with the Y215[5.75]A or the 210-215 deletion in GPR84 (Fig. 6c). It was speculated that the more extended TM5 could form additional interactions with the Ras domain of the Gαi protein[37,46], which would contribute to the extra pulling force and stabilize Gαi in a new energy-favorable orientation in the GPR84-Gαi complex, as shown in the SP1R-Gαi and CB2-Gαi complexes[37,46]. Indeed, the Gαi-coupled GPR84 structure here has the longest extended TM5 to our knowledge (Supplementary Fig. 9b), as shown by aligning these available receptor-Gαi protein complex structures using the GPR84 structure as a reference[46–56].

The C-terminal α5 helix of Gαi inserts into the cytoplasmic cavity of the receptor formed by TM3, TM5, TM6, and TM7 (Fig. 6a, d), which forms the main interaction interface, as observed in other Gαi-coupled GPCR structures[24,42,46,51,54−57]. On one side of the α5 helix, there is a hydrophobic network enhancing the interaction with GPR84, which is composed of I344, L348, C351, and L353 from Gαi, with L121[3.53] and I122[3.54] of TM3, I201[5.61] and V202[5.65] of TM5, and V317[6.33] and M320[6.36] of TM6 from GPR84 (Fig. 6d). On the other side of the α5 helix, the side chain of the C-terminal F354 of the α5 helix forms π-π stacking with the F314[6.30] of TM6 (Fig. 6e), and the terminal carboxyl group forms polar interactions with K316[6.32] of TM6 and R375[8.48] of TM8. At the top of α5, the main chain carbonyl group of C351 forms a hydrogen bond with R118[3.50] of the conserved DRY motif (GRY in GPR84), which is stabilized by a hydrogen bond of Y198[5.58] with R118[3.50] (Fig. 6e and Supplementary Fig. 8).

It has been accepted that the ICLs play a vital role in G protein coupling or selectivity, particularly ICL2[46,58]. Owing to the absence of interactions with ICL1 or ICL3, we moved our focus to the ICL2 region in the GPR84 structure, which folds into a short α-helix. The α-helix of ICL2 is highly compact because of the strong hydrophobic packing contributed by residues between TMD and ICL2, including F55[2.39], L113[3.45], L116[3.48], L120[3.52], L121[3.53], and I137[4.42] in TM2-4 and F132[34.57] and F128[34.53] from ICL2 (Fig. 6f), which causes a significant upward shift in the conformation of ICL2. Accordingly, ICL2 forms weak interactions with Gαi compared with other Gαi-coupled GPCR structures[37,47–56], with the only interaction formed between the main chain carbonyl group of D193 from Gαi and the side chain of K126[34.51] (Fig. 6g). Replacing K126[34.51] with alanine (phenylalanine or leucine) to disrupt this polar interaction reduced the maximum activation to about 86% of the wild-type receptor (Fig. 6c and Supplementary Table 2), suggesting a functional role of this position in GPR84-Gαi coupling. In a previous Gαi coupling study[57], the sequence analysis of amino acids at position 34.51 of all class A Gαi/o-coupled receptors revealed that most promiscuously Gαi/o-coupled receptors contain mostly very large/large hydrophobic or aromatic ring-containing amino acid residues, whereas primarily Gαi/o-coupled receptors also contain other residues, such as Ala, His, Pro, Ser/Thr, Arg/Lys, and Gln. To determine the exact roles of K126[34.51] and ICL2 in GPR84 signaling, we further performed sequence modifications of ICL2 and evaluated the Gαi- and Gαs-coupling activities of mutated GPR84 receptors.

Similar to the alanine substitution of K126[34,51], all these modifications of ICL2 displayed minor effects on the $G\alpha_i$-coupling of GPR84 and no enhancement of $G\alpha_s$ coupling (Supplementary Fig. 10), possibly due to the unique upward-shifted conformation of ICL2 in GPR84. In summary, we found conserved and divergent features in $G\alpha_i$ coupling to GPR84.

## Discussion

Structural studies of GPCRs, combined with molecular studies, have provided essential insights into ligand binding and GPCR activation mechanisms in the past decade. X-ray crystallography is a powerful tool to interpret the atomic details of GPCRs. Further assisted by the development of cryo-EM, tremendous progress has taken place with unprecedented efficiency, clarifying several innovative GPCR activation mechanisms, including the self-activation mechanism of GPR52, 5-HT$_{1A}$R[42,59], the beta-arrestin coupling mechanism[60–62], and the GRK recruitment mechanism[63,64]. In addition, cryo-EM structural studies of GPCRs also offer an alternative solution to G protein-coupled receptor deorphanizations, which is more direct and efficient for specific receptors[65]. In this paper, we obtained a MCFA-bound GPR84 structure, although no ligand was added during cryo-EM sample preparation. By extensive functional assays and additionally structural works, we modeled 3-OH-C12 as a template to study the activation of GPR84 by MCFAs, which could be accepted as putative endogenous activators of GPR84.

Our structures provide the amphiphilic orthosteric pocket of GPR84 and the "polar group out"-binding mode for both LY237 and 3-OH-C12. The pocket is principally located in the region between ECL2 on the extracellular side and the highly conserved residue position 6.48 toward the intracellular side. The NT-ECL2 disulfide bond in GPR84, together with the conserved disulfide bond between C93[3.25] and C168[ECL2], forms a unique disulfide bridge combination in GPR84 and governs the extracellular domain packing of ECL2 and the N-terminus for ligand binding and receptor activation. The molecular dynamics simulations and functional data reveal the binding pathway of LY237 through the ECL2-opening process. R172[ECL2] not only forms extensive interactions with the polar heads of both ligands in our structures but also guides ligand entry in our simulations with the aid of R349[ECL3] and H352[7.35]. The important role of R172[ECL2] in ligand accommodation for GPR84 was also validated by a plenty of molecular stimulations and functional assays[17,66], in which the proposed orientation of R172[ECL2] is accordant with that in our solved GPR84 structures. Both previous reports[9,15] and our functional data reveal that the potency of a GRP84 ligand is highly sensitive to the length of the alkyl tail. Analysis of these two structures reveals that the unique hydrophobic patch formed by F152[4.57] and L182[5.42] not only packs tightly with the alkyl tail of the MCFA-like agonist but also restricts its length, which helps to define the length of the binding pocket in GPR84. Our structure and function analysis also revealed the structural basis for the 270,000-fold greater potency of LY237 than 3-OH-C12, which accounts for the stronger polar and pi-packing interactions coordinating the polar end of LY237, including the participation of S169[ECL2], R172[ECL2], F335[6.51], and W360[7.43] in the LY237 structure.

Lipid-binding GPCRs play important physiological roles, accordingly, they are attractive targets for the treatment of many pathophysiological conditions. Our solved structures of GPR84 here reveal key residues that interact with two kinds of agonists with different polar ends, which could enable structure-based drug design to facilitate the discovery of more powerful ligands. In addition, the structures reveal that ECL2, with a significant downward-shifted conformation in GPR84, contributes to ligand binding and plays a pivotal role in ligand entry from the solvent. Together, these studies provide the structural basis for understanding the ligand recognition and activation of GPR84 as well as structural templates for the rational design of small molecule drugs targeting GPR84.

## Methods

### Protein complex expression and purification

The wild-type human GPR84, modified with an N-terminal thermally stabilized BRIL as a fusion protein, along with a Flag tag and His tag epitope, was cloned into pFastBac transfer plasmid (Invitrogen). All constructs were generated using the Phanta Max Super-Fidelity DNA Polymerase (Vazyme Biotech Co.,Ltd) and verified by DNA sequencing (Genewiz). A dominant-negative (DN) $G\alpha_i$ format including mutation G203A/A326S was constructed to increase the stability of $G\alpha_i\beta_1\gamma_2$ complex[67]. The NanoBiT tethering method was used to facilitate assembly of the GPR84-$G\alpha_i$ complex, in which the C-terminus of GPR84 was fused to the large part of NanoBiT (LgBiT), and the C-terminus of $G\beta$ was fused to the small part of NanoBiT (SmBiT)[20].

GPR84, $G\alpha_i$, $G_{\beta1}$, $G_{\gamma2}$ and scfv16 were co-expressed in *Trichoplusia ni* (Hi5) insect cells. Baculoviruses were prepared using the Bac-to-Bac Expression System (Invitrogen). Cells were infected with viruses at density of $2.8 \times 10^6$ cell/ml and cell culture was collected by centrifugation 48 h post-infection and stored at −80 degree until use.

For the purifications of LY237-bound GPR84-$G\alpha_i$ complex, cell pellets were thawed in 20 mM HEPES pH 7.4, 100 mM NaCl, 5 mM MgCl$_2$, 5 mM CaCl$_2$, and protease inhibitor cocktail (TargetMol.USA). The suspensions were incubated for 1 h at room temperature supplemented with 1 μM LY237 and 25 mU/ml apyrase (Sigma-Aldrich). Subsequently, 0.5% (w/v) n-dodecyl-β-d-maltoside (DDM, Anatrace) and 0.1% (w/v) cholesteryl hemisuccinate (CHS, Anatrace) were added to solubilize complexes for 2 h at 4 °C. Insoluble material was removed by centrifugation at $30,000 \times g$ for 30 min and the supernatant was immobilized by batch binding to Talon affinity resin. After that, the resin was packed and washed with 20 column volumes of 20 mM HEPES pH 7.4, 100 mM NaCl, 1 μM LY237, 20 mM imidazole, 0.01% (w/v) LMNG, and 0.002% (w/v) CHS. Finally, the complex was collected in buffer containing 250 mM imidazole and concentrated using an Amicon Ultra Centrifugal Filter (MWCO 100 kDa). Complexes were loaded onto a Superdex 200 Increase 10/300 column (GE Healthcare) with buffer containing 20 mM HEPES pH 7.4, 100 mM NaCl, 1 μM LY237, 0.00075% (w/v) LMNG, 0.0002% (w/v) CHS to separate complex from contaminants. Eluted fractions consisting of GPR84-$G\alpha_i$ complex were pooled and concentrated for electron microscopy experiments. For the purifications of no exogenous ligand added GPR84-$G\alpha_i$ complex and 3-OH-C12 bounded GPR84-$G\alpha_i$ complex, no ligand and 200uM 3-OH-C12 were added through every step of purification.

### Cryo-EM grid preparation and data collection

For cryo-EM grids preparation of the GPR84-$G\alpha_i$ complexes, 3 μl of the protein at ~20 mg/ml were loaded onto a glow-discharged holey carbon grid (Quantifoil Au 300 mesh R1.2/1.3), and subsequently were plunge-frozen in liquid ethane using Vitrobot Mark IV (Thermo Fischer Scientific). Cryo-EM imaging was collected on a Titan Krios at 300 kV using Gatan K3 Summit detector with a pixel size of 0.824 Å at the Shanghai Advanced Center for Electron Microscopy, Shanghai Institute of Materia Medica, Chinese Academy of Sciences. Images were taken at a dose rate of about 8.0 e/Å²/s with a defocus ranging from −1.0 to −2.0 μm using the EPU software (FEI Eindhoven, Netherlands). The total exposure time was 8 s, and 36 frames were recorded per micrograph. A total of 15,833 and 11,844 movies were collected for LY237-bound and no ligand bound GPR84-$G\alpha_i$ complexes, respectively. For 3-OH-C12-GPR84-$G\alpha_i$ complex, Cryo-EM imaging was collected on a Titan Krios equipped with a Falcon 4 direct electron detection device at 300 kV. Images were taken with a pixel size of 0.73 Å, a defocus ranging from −1.0 to −2.0 μm using the EPU software (FEI Eindhoven, Netherlands). We collected a total of 9816 movies with total dose of 50 e Å$^{-2}$ s$^{-1}$ over 2.5 s exposure on each EER format movie. Each movie was divided into 36 frames during motion correction.

## Image processing and map construction

The single particle analysis of GPR84-Gα$_i$ complexes was performed with cryoSPARC v3.3[68]. Dose-fractionated image stacks were subjected to motion correction by MotionCor2.1[69]. Contrast transfer function (CTF) parameters for micrograph were estimated by patch CTF estimation. For the LY237-GPR84-Gαi complex, a total of 11,368,094 particles were auto-picked from 15,833 micrographs. The map of D2R-Gα$_i$ complex[43] (EMD-22511) was used for reference-based template-matching. The picked particles were extracted on a binned dataset with a pixel size of 1.648 Å and were subjected to reference-free 2D classification. The datasets with well-define averages were re-extracted with pixel size of 0.824 and were subjected to interactive 2D and 3D classifications. After multiple rounds of hetero-refinement, a total of 373,108 particles were subsequently subjected to non-uniform refinement, local refinement. The final density map was obtained with an overall resolution of 3.23 Å at a Fourier shell correlation of 0.143. For the GPR84-Gα$_i$ complex with no ligand modeled, a total of 9,503,626 particles were auto-picked, extracted on a binned dataset with a pixel size of 1.648 Å, and subjected to 2D classification. The datasets with well-define averages were re-extracted with pixel size of 0.824 and were subjected to interactive 2D and 3D classifications. A total of 260,582 particles were selected and subjected to non-uniform refinement, local refinement. The final density map was obtained with a nominal resolution of 3.24 Å at a Fourier shell correlation of 0.143. For 3-OH-C12 bounded GPR84-Gα$_i$ complex, a total of 5,647,309 particles were auto-picked, extracted on a binned dataset with a pixel size of 1.46 Å, and subjected to 2D classification. The datasets with well-define averages were re-extracted with pixel size of 0.73 and were subjected to interactive 2D and 3D classifications. A total of 213,390 particles were selected and subjected to non-uniform refinement, local refinement, and deep EMhancer. The final density map was obtained with a nominal resolution of 2.89 Å at a Fourier shell correlation of 0.143.

## Model building and refinement

Predicted model of active-state GPR84 receptor from Alfa-fold were used as initial model for rebuilding and refinement against the electron microscopy density map. UCSF Chimera-1.14[70] was used to dock the model into the electron microscopy density map, and followed by iterative manual adjustment and rebuilding in COOT-0.9.6[71] and ISOLDE-1.2[72]. Then models were further refined and validated in Phenix-1.20[73] programs (Supplementary Table 3). Structural figures were generated using UCSF Chimera-1.14, ChimeraX-1.2[74] and PyMOL-2.0 (https://pymol.org/2/).

## Mass spectrometry

The GPR84-Gi complex was placed into a 1.5 ml centrifuge tube after thawing in ice/water. 200 μl methanol/water (75:25, v/v) solvent were added and mixed for 2 min. Then 0.5 ml MTBE was added, vortexed for 2 min, and shacked for 30 min, and the samples were homogenized by ultrasonic for 5 min. Subsequently, 130 μl water was added and mixed for 2 min. The mixture was centrifuged at 14,000 × *g* for 20 min at 4 °C. About 200 μl of the supernatant was taken out and frozen to dry. The dried sample was redissolved in 60 μl acetonitrile/isopropanol/water (65:30:5, v/v/v) for LC-MS analysis. The sample was analyzed on an Agilent 6545 Q-Tof LCMS system coupled with MassHunter Workstation Data Acquisition software in the positive and negative ion modes. A Waters ACQUITY UPLC BEH C8 column (100 × 2.1 mm, 1.7 μm) was employed for separation at a flow rate of 0.26 ml/min at 55 °C. Mobile phase A consists of acetonitrile: water (60:40, v/v) with 10 mM ammonium formate, and mobile phase B consists of isopropanol: acetonitrile (90:10, v/v) with 10 mM ammonium formate. The gradient elution was set as follows: 0-1.5 min 32% B; 1.5-15.5 min 85% B; 15.5-15.6 min 97% B; 15.6-18 min 97% B; 18.1-21 min 32% B. The mass spectrum parameters were set as follows: mass range 100-1500 *m/z*; gas temperature 300°C; drying gas 9 l/min; sheath gas flow 12 l/min; framentor 135 V; acquisition rate 10 spectra/s. MS/MS data were acquired on top 10 max precursor, and collision energy was set at 25,35,45 V. The peak extraction and lipids identification were achieved by MSDIAL (v4.8).

## cAMP detection assay

For detecting Gα$_i$ signaling of GPR84, HEK293 cells expressing WT or mutant GPR84 were harvested and re-suspended in DMEM containing 500 μM IBMX at a density of $2 \times 10^5$ cells/ml. Cells were then plated onto 384-well assay plates at 2000 cells/5 μl/well. Another 5 μl buffer containing 3 μM Forskolin and various concentrations of test compounds, such as LY237 and C8-C20 (Targetmol.USA) were added to the cells. After incubation at room temperature for 15 min, intracellular cAMP level was tested by a LANCE Ultra cAMP kit (PerkinElmer, TRF0264) and EnVision multiplate reader according to the manufacturer's instructions. All functional data are processed in GraphPad Prism 8.0.

## Gα$_i$ and Gα$_s$ recruitment assay

The recruitment of Gα$_i$/Gα$_s$ to GPR84 was measured using the Promega NanoBiT Protein-Protein Interaction System. In brief, HEK293 cells seeded at $5 \times 10^4$ cells/well on 96-well plates were co-transfected with plasmids encoding GPR84-LgBiT and SmBiT-mini Gα$_i$ or Gα$_s$ at the ratio of 1:1. Twenty-four hours later, cells were replaced with 40 μl of fresh culture medium (without FBS). And then added 10 μl Nano-Glo Live Cell reagent according to the manufacturer's protocol (Promega, Cat No: N2011), incubated in a 37 °C, 5% CO$_2$ incubator for 10 min. Another 25 μl culture medium with various concentrations of compounds were added to the cells. After incubation at room temperature for 10 minutes, bioluminescence was measured with an EnVision multiplate reader (PerkinElmer). All functional data are processed in GraphPad Prism 8.0.

## Surface expression analysis

Cell-surface expression for WT GPR84 and mutants was monitored by a fluorescence-activated cell sorting (FACS) assay. In brief, HEK293 cells expressing HA-tagged GPR84 were harvested twenty-four hours after transfection. Cells were suspended by PBS buffer with 5 mM EDTA and blocked with 5% (w/v) BSA at room-temperature for 30 min, then followed by the incubation with mouse anti-HA-FITC antibody (Sigma) at a dilution of 1:200 for 30 min at 4 °C, and then PBS buffer was added to cells. Finally, the surface expression of GPR84 was monitored by detecting the fluorescent intensity of FITC with Guava easyCyte 8HT (Merck Millipore). The FACS data were analyzed by Guava software 2.1.

## MD simulations

The cryo-EM structure of GPR84-LY237-Gα$_i$ complex was used for the construction of MD simulation system. Before the simulation, Gα$_i$ was removed from the system. 10 LY237 were placed out of the complex for potential ligand entry process. We oriented the structures locally and used CHARMM-GUI server to insert them into POPC (palmitoyl-2-oleoyl-sn-glycero-3-phosphocholine) membrane. Then, TIP3P waters were added on the top and bottom of the system. In all, 0.15 mol/l NaCl ions and counterions were finally added to solvent[75,76]. FF19SB, LIPID17, and GAFF2 force field were applied for the parameter of amino acids, lipids, and LY237, respectively[77,78].

The following simulation processes were finished on Amber20. Firstly, the system encountered a minimization process of 5000 cycles with a constraint on the backbone atom, sidechain atom, and lipid coordinates and constraint on dihedrals. Then, the constraints were generally decreased in the separated 6 steps of the equilibration process provided by CHARMM-GUI. Next, paralleled 8 times of LiGaMD running was applied on the system.

**Table 1 | Parameters for non-bonded potential energy boosting**

| Repeat | $V_{max}$ (kcal/mol) | $V_{min}$ (kcal/mol) | $V_{avg}$ (kcal/mol) | $\sigma_V$ (kcal/mol) |
|---|---|---|---|---|
| 1 | 43.0303 | −98.2983 | −29.7762 | 7.9339 |
| 2 | 39.9781 | −92.442 | −38.1154 | 8.856 |
| 3 | 128.9275 | −95.8721 | −31.8457 | 7.4627 |
| 4 | 123.1266 | −92.8427 | −33.1884 | 7.9531 |
| 5 | 32.2203 | −88.7339 | −33.5473 | 8.6824 |
| 6 | 53.8041 | −101.869 | −40.5985 | 8.3789 |
| 7 | 97.4856 | −95.6821 | −27.5663 | 6.0549 |
| 8 | 88.0037 | −90.6698 | −31.9644 | 7.6115 |

**Table 2 | Parameters for remaining potential energy boosting**

| Repeat | $V_{max}$ (kcal/mol) | $V_{min}$ (kcal/mol) | $V_{avg}$ (kcal/mol) | $\sigma_V$ (kcal/mol) |
|---|---|---|---|---|
| 1 | −188,193.0264 | −195,099.5019 | −188,535.2496 | 63.9921 |
| 2 | −188,122.5559 | −195,222.3564 | −188,501.6646 | 64.5597 |
| 3 | −186,760.9133 | −195,289.0602 | −187,295.3250 | 69.3229 |
| 4 | −187,293.8718 | −195,040.6451 | −187,751.5840 | 66.7513 |
| 5 | −188,080.277 | −195,056.8473 | −188,440.3333 | 64.7048 |
| 6 | −187,432.2234 | −195,201.6755 | −187,889.6753 | 67.3306 |
| 7 | −187,140.8344 | −195,235.8072 | −187,596.0806 | 67.3265 |
| 8 | −186,282.0971 | −195,124.0269 | −186,878.0558 | 68.9874 |

In each process, 1.2 ns conventional MD was first applied to the system to further equilibrate the system. During the 1.2 ns simulations, potential energies were not collected for calculating their LiGaMD parameters. Then, 12 ns conventional MD was used to calculate LiGaMD acceleration parameters, including the non-bonded potential energy of the bound ligand ($E_{nb}$) and remaining potential energy of the rest of the system ($E_{rp}$). Then, the boost potential $\Delta V$ for non-bonded and potential energy was calculated by Eq. (1).

$$\Delta V = \begin{cases} \frac{1}{2} k_0 (E_{th} - V)^2, & V < E_{th} \\ 0, & V > E_{th} \end{cases} \quad (1)$$

where $k_0$ was the harmonic constant and $E_{th}$ was the threshold energy. For non-bonded potential energy, $k_0$ and $E_{th}$ were calculated by Eqs. (2) and (3), respectively.

$$k_0 = \left(1 - \frac{\sigma_0}{\sigma_V}\right) \frac{V_{max} - V_{min}}{V_{avg} - V_{min}} \quad (2)$$

$$E_{th} = V_{min} + \frac{V_{max} - V_{min}}{k_0} \quad (3)$$

For remaining potential energy, $k_0$ and $E_{th}$ were calculated by Eqs. (4) and (5), respectively.

$$k_0 = \min\left(1.0, \frac{\sigma_0}{\sigma_V} \times \frac{V_{max} - V_{min}}{V_{max} - V_{avg}}\right) \quad (4)$$

$$E_{th} = V_{max} \quad (5)$$

where $V_{max}$, $V_{min}$, $V_{avg}$, and $\sigma_V$ are the maximum, minimum, average and standard deviation of the corresponding potential energy value, and $\sigma_0$ was 6 kcal/mol as guided in Amber tutorial. These values were used during the following 1.2 ns LiGaMD equilibrium runs. Then 60 ns parameter sampling were applied for updating these potential energy values under LiGaMD condition. The parameters for each run are listed

in Tables 1 and 2. During the 73.2 ns sampling runs, all the parameters have converged as in Supplementary Figs. 11 and 12. 1000 ns LiGaMD production simulation was finally performed on each system. We applied the "dual-boost" bias potential to the system. When the distance between the NE1 atom at W360[6.48] and the O17 atom at LY237 was more than the distance in the cryo-EM structure (4.2 Å), the boost potential was applied to the ligand non-covalent interaction potential energy with GPR84 to accelerate ligand entry. Meanwhile, the potential energy of the entire system was also boosted to accelerate sampling. From the last structure of the representative ECL2 open trajectory whose ligand was in the pocket, we continuously run a 1000 ns conventional MD simulation to observe the ligand behavior.

During simulations, the temperature (310 K) and pressure (1 atm) were controlled by Langevin thermostat and isotropic Berendsen barostat, respectively. Long-range electrostatic interactions were treated by Particle mesh Ewald algorithm and a cutoff of 9 Å was employed for short-range interactions. The SHAKE algorithm was applied to restrain the bond with hydrogens, so the timestep of simulations was 2 fs. The analyses of simulation trajectories were finished by CPPTRAJ[79].

### Reporting summary
Further information on research design is available in the Nature Portfolio Reporting Summary linked to this article.

## Data availability
Density maps and structure coordinates have been deposited in the Electron Microscopy Data Bank (EMDB) and the Protein Data Bank (PDB) with accession codes EMD-35913 and 8J18 for 3-OH-C12-GPR84-Gα_i complex; EMD-35914 and 8J19 for LY237-GPR84-Gα_i complex; and EMD-35915 and 8J1A for GPR84-Gαi complex with no ligand modeled. The mass spectrum analyses on the GPR84-Gα_i complex with no ligand supplied were supplied in Supplementary Data 1. Source data are provided with this paper.

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

## Acknowledgements

The cryo-EM data of the GPR84 samples were collected at the Shanghai Advanced Electron Microscope Center, Shanghai Institute of Material Medica. We thank all the staff (Qingning Yuan, Kai Wu, Wen Hu, and Shufeng Zhang) at the cryo-EM facility for their technical support. We also thank Hu Zhou and Pengjun Cai at the Department of Analytical Chemistry, Shanghai Institute of Material Medica, for their technical support in our MS assay. This work was partially supported by CAS Strategic Priority Research Program (XDB37030103 to H.E.X.); Shanghai Municipal Science and Technology Major Project (2019SHZDZX02 to H.E.X.); Shanghai Municipal Science and Technology Major Project (H.E.X.); The National Natural Science Foundation of China (82121005 to X.X. and H.E.X., 32130022 to H.E.X., 81730099 to X.X, 32171189 to W.Y.); the National Key R&D Program of China (2018YFA0507002 to H.E.X.); the Youth Innovation Promotion Association of CAS (2020283 to Q.Z., 2021278 to W.Y.); National Science Fund for Excellent Young Scholars (82122067 to W.Y.); Key tasks of the Lingang Laboratory (LG202103-03-05 to W.Y.). In addition, this work was partially supported by High-level new R&D institute (2019B090904008), and High-level Innovative Research Institute (2021B0909050003) from Department of Science and Technology of Guangdong Province, and the author W.Y. also gratefully acknowledges the support of Sanofi Scholarship Program.

## Author contributions

H.L. designed the expression constructs, purified the complexes, performed cryo-EM grid preparation, data collection and procession, model building and refinement, and participated in figure and manuscript preparation; Q.Z. performed plasmid construction and functional studies, participated in figure and manuscript preparation, with the help of S.W. and X.Y.; X.C. supervised X.H., performed MD studies, participated in figure and manuscript preparation; M.J. participated in plasmid construction and protein purification; F.-J.N. supervised Y.L., synthesized the LY237; X.X. supervised the functional studies; H.E.X., X.X., and W.Y. conceived, designed and supervised the overall project. H.E.X., X.X., W.Y., H.L., Q.Z., and X.H. participated in data analysis and interpretation, and wrote the manuscript with inputs from all authors.

## Competing interests

The authors declare no competing interests.
