## [Peer Review File · Nature Communications]

Structural insights into ligand recognition and activation of the medium-chain fatty acid-sensing receptor GPR84REVIEWER COMMENTS

Reviewer #1 (Remarks to the Author):

Liu et al. reported cryo-EM structures of the GPR84-Gi complex bound to LY237 or 3-OH-C12. This study described the unique ligand-binding pocket of GPR84 and proposed a possible ECL2-guided ligand entry mechanism. Furthermore, this work identified 3-OH-C12 as an endogenous agonist contributing to the de-orphan of GPR84. In addition, comparisons with other lipid-binding GPCRs uncover the similar receptor activation mechanism and identical G protein coupling mode of GPR84, highlighting the differences in ligand recognition. Therefore, the results from this study will potentially be of considerable interest to the field of lipid-activated GPCR structures and mechanisms and exciting to those making anti-inflammation drug development efforts. Overall, the data presented here is of good quality (e.g., the overall quality of cryo-EM data and the pharmacological data), only a few issues should be addressed.

1. I appreciate the authors interrogating the efficacies of different FFAs activating GPR84. However, it is not easy to conclude that the omit map is 3-OH-C12. Since the complex is purified without an exogenous compound, the endogenous ligand comes from the cells or medium. Is 3-OH-C12 abundant in cells and physiologically important? The authors should consider other strategies (e.g., mass spectrometry) to prove the existence and relative abundance of 3-OH-C12 in the GPR84-Gi complex.

2. The cryo-EM map and the model for the 3-OH-C12 bound state are relatively low-quality, leading to the ambiguity of the modeled molecule. The authors did functional interrogation of the pocket residues with the cAMP assay, but the results still do not break the ambiguity in the ligand binding pose. Could the authors improve the current cryo-EM map quality by further processing or collecting additional data and refining the model properly? Furthermore, why not directly assemble the complex by adding exogenous 3-OH-C12 or other FFAs for structure determination?

3. Similarly, the EM map cannot help the authors to fix the orientation of the pyridine ring in LY237. According to the analysis of LY237 binding, the nitrogen on the pyridine ring has a negligible effect. Interestingly, upon binding to the receptor, the pyridine ring rotates 180 degrees in some snapshots in the MD video. If a benzyl ring could replace the pyridine? Is there any possibility to further improve the potency or specificity of LY237 by strengthening the interaction in this part?

4. Given the importance of the N-terminus and ECL-2 loop conformations, please provide the density map of the N-terminus and ECL-2 loop in the figure to better present the interactions.

5. Some minor points: a few panels and figure legends should be adjusted. For example,

(a) The space-filling representations of disulfide bonds shown in Fig. 1c and S3 are a little bit crowded and better to change to sticks;

(b) The legends for Fig 2b-d are confused.

(c) The line styles and colors of Fig 3f are inconsistent with the description in legends;

(d) The residue label in Fig 5d is missing;

Reviewer #2 (Remarks to the Author):

In this manuscript, Liu et al., have obtained and characterized the cryoEM structure of GPR84 bound to LY237 or 3-OH-C12 agonists in the presence of a Gi protein. The structure is supported by the wealth of mutagenesis data and explored by molecular dynamics simulations. From this integrated approach, the authors provide the unique features of a receptor structure, ligand binding pathway and interactions, receptor activation, and G-protein coupling. The manuscript is of a high standard, but several issues should be addressed before its publication:

1. The section on 3-OH-C12. The authors should elaborate more on how they obtained the structure of GPR84 bound to 3-OH-C12. The current text is not convincing that they tried to get an empty receptor but obtained the receptor bound to 3-OH-C12 by an accident not adding the ligand. How comes such a rare lipid which is not typical for a lipid continuum used could sit in the protein mixture for cryoEM?

2. Line 251, 'The closed ECL2 in the cryoEM structure spontaneously opened in 3 of 8 LiGaMD simulations' please clarify here, the loop opens spontaneously in the absence of the ligand nearby in three simulations, and other five it opens when the ligand comes close and under influence of the ligand. I assume that the loop should be opened in all the simulations during the entrance process otherwise the ligand cannot enter.

3. Please show the distances (h_{ligand} , h_{ECL2}) that were monitored in MD simulations in Figure 4.

4. MD simulations methods. Please specify the values of a boost potential to bring the ligand inside the receptor and to enhance the sampling of the receptor conformations, and how these values were selected, and convergence achieved.

5. Line 256. 'Mutations to disrupt these interactions' What mutations do authors mean? Mutations of V164, C166 or C11. Please specify.

6. Lines 419-420. The disulfide bridge between C11 and C166 has been predicted by AlphaFold and used to explain the SAR of triazine antagonists. Please make references to AlphaFold and Investigating the Structure-Activity Relationship of 1,2,4-Triazine G-Protein-Coupled Receptor 84 (GPR84) Antagonists.

Mahindra A, Jenkins L, Marsango S, Huggett M, Huggett M, Robinson L, Gillespie J, Rajamanickam M, Morrison A, McElroy S, Tikhonova IG, Milligan G, Jamieson AG. *J Med Chem.* 2022 Aug 25;65(16):11270-11290.

7. Lines 424-426. The anchoring role of R172 for agonist binding has been previously described and the cryoEM structure confirms its direct involvement in coordinating the agonist. Please cite two references: Tikhonova I.G. Application of GPCR structures for modelling of free fatty acid receptors. *Handb. Exper. Pharmacol.* 2017;236:57–77. Mahmud Z.A., Jenkins L., Ulven T., Labéguère F., Gosmini R., De Vos S., Hudson B.D., Tikhonova I.G., Milligan G. Three classes of ligands each bind to distinct sites on the orphan G protein-coupled receptor GPR84. *Sci. Rep.* 2017;7:17953.

8. Line 426. "Previous reports' please add the references.

9. Video: please visualise the key polar residues (R349, H352 and R172) stabilizing various states during the binding process.

10. Line 249, applied ->used.

11. Line 253, indicating an opening tendency -> remove.

Reviewer #3 (Remarks to the Author):

This manuscript by Liu et al. reported 3.2 Å-resolution Cryo-EM structure of GPR84•Gi complex, which was reported to be involved in immunity by activating macrophages. While several Cryo-EM lipid-sensing GPCR structures coupled with Gi have already been reported, such as S1PR and LPA1 receptors, and activation mechanisms are similar, the outstanding progress in this manuscript is ligand entry mechanism involving lid opening revealed by MD simulation and complementary functional analysis. First, the authors solved the GPR84 complex structure without any ligand nor agonist, they found EM density in the orthosteric pocket, which resembles that of MCFAs such as 3-OH-C12, a putative endogenous ligand for GPR34. Thus they solved the activated structure of GPR34 in complexes with an endogenous ligand and super agonist, LY237. Intermediate structures of MD simulation were confirmed by functional analysis, which is very original and consistent with the simulation. The GPCR activation mechanism as well as Gi interaction mechanism are almost similar to the other class-I GPCRs. The structural analysis is solid and the paper is well written. Therefore, this reviewer recommends publication of this paper in *Nature Communications*, once the authors properly address the following minor comments:

1. As the LY237 complex structure is the main story of this manuscript, the author should present structural formula of LY237 in main Fig. 1.

2. In line 164, the authors should cite reference of LPA1 (2 papers) and S1PR.

3. In lines 272-273, there are some typo, since the following residue numbering is inconsistent with Fig. 4 ?; V165 should be V164 and A356 should be A365.

Manuscript ID: Nature Communications Manuscript (NCOMMS-22-49620-T)

Title: Structural insights into ligand recognition and activation of the medium-chain fatty acid-sensing receptor GPR84

We want to thank the three reviewers for their tremendous efforts in evaluating our manuscript and for their positive comments on the importance and biological significance of the ligand recognition and activation of the medium-chain fatty acid-sensing receptor GPR84. Their constructive suggestions have helped us tremendously in revising our manuscript. In response to their comments, we have performed additional experiments, including mass spectrometry analysis of the GPR84-Gi complex without exogenous ligands applied, and structural determination of the GPR84 complex activated by our conjectured 3-OH-C12 at a better resolution of 2.89 Å. The 3-OH-C12 density in our new structure matches well with our previous model determined without exogenous ligands applied, further validating the 3-OH-C12 as a putative endogenous ligand for GPR84 function. In addition, we also performed molecular docking of a panel of agonists with different skeleton structures including analogs of both LY237 and 3-OH-C12, and provided the associated SAR data. We think that the revised paper is much improved with better quality and resolution of our structures. In the following sections, we provide point-by-point responses to the comments by the three reviewers of our original paper. The reviewer's comments are in **black** and our responses are in **blue**.

Point-by point response to Reviewer #1:

Reviewer #1: Liu et al. reported cryo-EM structures of the GPR84-Gi complex bound to LY237 or 3-OH-C12. This study described the unique ligand-binding pocket of GPR84 and proposed a possible ECL2-guided ligand entry mechanism. Furthermore, this work identified 3-OH-C12 as an endogenous agonist contributing to the de-orphan of GPR84. In addition, comparisons with other lipid-binding GPCRs uncover the similar receptor activation mechanism and identical G protein coupling mode of GPR84, highlighting the differences in ligand recognition. Therefore, the results from this study will potentially be of considerable interest to the field of lipid-activated GPCR structures and mechanisms and exciting to those making anti-inflammation drug development efforts. Overall, the data presented here is of good quality (e.g., the overall quality of cryo-EM data and the pharmacological data), only a few issues should be addressed.

Response: We are very grateful for the positive assessment on the quality and importance of our work by the Reviewer.

Reviewer #1: 1. I appreciate the authors interrogating the efficacies of different FFAs activating GPR84. However, it is not easy to conclude that the omit map is 3-OH-C12. Since the complex is purified without an exogenous compound, the endogenous ligand comes from the cells or medium. Is 3-OH-C12 abundant in cells and physiologically important? The authors should consider other strategies (e.g., mass spectrometry) to prove the existence and relative abundance of 3-OH-C12 in the GPR84-Gi complex.

Response: We thank the reviewer for the suggestion. Firstly, the medium manufacturer we consulted told us that the medium was free of medium chain fatty acids or their derivatives, so

it is likely that the bound ligand solved in our structure comes from the insect cells or their metabolites. To determine the exact identity of the bound ligand, we performed mass spectrum (MS) analyses on the protein sample used for our structural studies. However, only fatty acid with carbon length of more than 14 carbon atoms is identified, without any short- or medium-chain fatty acids with 3-OH substituents (Table. R1). The substrate of the density may come from degradation of these fatty acids. In fact, it is difficult to identify the accurate lipid or fatty acid composition in the GPR84-G_i complex purified from our insect cell protein expression system, which is partially due to the complex and dynamic features of fatty acids or their derivatives in eukaryotic cell. This is accordant with our mass spectrometry analysis of the GPR84-G_i complex without exogenous ligands applied.

1	Alignment	Average Rt	Average Mz	Metabolite name	Adduct type	blank1-n	blank2-n	GPr84-1-n	GPr84-2-n
2	659	8.649	596.52582	Cer 34:1,2O	[M+CH3COO]-	4309	3890	643184	804825
3	900	7.649	761.58099	Cer 34:1,2O	[M+CH3COO]-	0	0	68511	57720
4	701	9.56	624.55737	Cer 36:1,2O	[M+CH3COO]-	1040	3357	349130	445616
5	743	10.234	652.58838	Cer 38:1,2O	[M+CH3COO]-	411	0	33911	40377
6	744	10.413	652.58844	Cer 38:1,2O	[M+CH3COO]-	807	722	32485	31716
7	135	1.846	227.20195	FA 14:0	[M-H]-	71796	57948	256008	367762
8	132	1.619	225.18575	FA 14:1	[M-H]-	7622	2852	100440	121283
9	210	2.369	255.23605	FA 16:0	[M-H]-	2952894	2441504	6035324	6461931
10	201	1.986	253.22089	FA 16:1	[M-H]-	51823	31256	5853582	6740062
11	198	1.694	251.20216	FA 16:2	[M-H]-	1482	559	44223	56531
12	217	2.706	269.24817	FA 17:0	[M-H]-	28838	20566	10563	15635
13	215	2.235	267.23224	FA 17:1	[M-H]-	5999	2851	25690	33165
14	242	3.178	283.26736	FA 18:0	[M-H]-	1608338	1735419	5548878	6167828
15	230	2.549	281.25461	FA 18:1	[M-H]-	170720	75936	8856731	9718514
16	224	2.099	279.23376	FA 18:2	[M-H]-	210120	92519	710004	1128552
17	222	1.768	277.21652	FA 18:3	[M-H]-	16595	8579	45663	67422
18	290	4.28	311.29562	FA 20:0	[M-H]-	10199	10415	304312	448058
19	285	3.391	309.27939	FA 20:1	[M-H]-	2924	308	129742	190566
20	332	5.537	339.32675	FA 22:0	[M-H]-	1480	637	241847	339633
21	366	6.773	367.35748	FA 24:0	[M-H]-	1878	787	22495	31636
22	474	1.987	452.27835	LPE 16:0	[M-H]-	0	0	264809	284016
23	472	1.672	450.26205	LPE 16:1	[M-H]-	0	0	60433	66594
24	508	2.571	480.30939	LPE 18:0	[M-H]-	0	0	508745	481220
25	504	2.11	478.29416	LPE 18:1	[M-H]-	0	0	1151010	1245103
26	542	3.492	508.34186	LPE 20:0	[M-H]-	0	0	52106	40132
27	907	7.38	764.54205	PC 30:0	[M+CH3COO]-	0	0	8969	11379
28	902	6.593	762.52698	PC 30:1	[M+CH3COO]-	0	0	23896	29759
29	924	6.796	788.54449	PC 32:2	[M+CH3COO]-	0	0	104425	111391
30	953	8.48	818.59204	PC 34:1	[M+CH3COO]-	0	0	151463	165568
31	951	7.762	816.57629	PC 34:2	[M+CH3COO]-	0	0	169093	198631
32	949	7.042	814.56207	PC 34:3	[M+CH3COO]-	0	0	7118	9565
33	969	9.313	846.62305	PC 36:1	[M+CH3COO]-	0	0	78712	86015
34	768	6.773	660.46143	PE 30:1	[M-H]-	0	0	17093	17447
35	816	8.48	690.50775	PE 32:0	[M-H]-	0	0	15440	11363
36	812	7.761	688.49237	PE 32:1	[M-H]-	0	0	301105	328982
37	807	6.975	686.47595	PE 32:2	[M-H]-	0	0	91457	99665
38	854	9.313	718.53687	PE 34:0	[M-H]-	0	0	6542	4434
39	848	8.66	716.52368	PE 34:1	[M-H]-	0	0	886406	737374
40	841	7.93	714.50757	PE 34:2	[M-H]-	0	0	710490	640214
41	891	9.469	744.55499	PE 36:1	[M-H]-	0	0	357392	226048
42	889	8.796	742.53943	PE 36:2	[M-H]-	0	0	1140942	1043732
43	885	8.121	740.52234	PE 36:3	[M-H]-	0	0	36046	34327
44	911	10.212	772.58777	PE 38:1	[M-H]-	0	0	19681	12593
45	928	8.683	789.61255	SM 36:1,2O	[M+CH3COO]-	0	0	32274	25574

Table. R1 Representative compound from our mass spectrum (MS) results. The boxed column indicates compounds identified by mass spec.

For the potentially functional role of 3-OH-C12, it is reported that the 3-OH-C12 is actually formed during β -oxidation in peroxisomes and its plasma concentration can reach micromolar order in mitochondrial fatty acid β -oxidation disorders (Ref-1 and Ref-2). Soydan et al. found that 3-OH-C12 can stimulate the release of IL-6, one of the key proinflammatory cytokines, from human blood cells in vitro (Ref-3). Our previous study has found GPR84 knockout or blockade significantly reduced IL-6 release from macrophages (Ref-4). These evidences also

support that the 3-OH-C12 is a bona fide endogenous MCFA, which plays a role in the energy metabolism, inflammation, and et al, most likely through binding and activation of GPR84.

References

Ref-1. Costa, C. G., Dorland, L., Holwerda, U., de Almeida, I. T., Poll-The, B. T., Jakobs, C., and Duran, M. (1998) Simultaneous analysis of plasma free fatty acids and their 3-hydroxy analogs in fatty acid β -oxidation disorders. *Clin. Chem* 44, 463–471

Ref-2. Jones, P. M., and Bennett, M. J. (2011) Clinical applications of 3-hydroxy fatty acid analysis by gas chromatography-mass spectrometry. *Biochim. Biophys. Acta* 1811, 657–662.

Ref-3. Soydan, A. S., Dokmetas, H. S., Cetin, M., Koyuncu, A., Kaptanoglu, E., and Elden, H. (2006) The evaluation of the role of β -hydroxy fatty acids on chronic inflammation and insulin resistance. *Mediators Inflamm.* 2006,64980

Ref-4. Zhang, Q. et al. GPR84 signaling promotes intestinal mucosal inflammation via enhancing NLRP3 inflammasome activation in macrophages. *Acta pharmacologica Sinica* 43, 2042-2054, doi:10.1038/s41401-021-00825-y (2022).

Reviewer #1: 2. The cryo-EM map and the model for the 3-OH-C12 bound state are relatively low-quality, leading to the ambiguity of the modeled molecule. The authors did functional interrogation of the pocket residues with the cAMP assay, but the results still do not break the ambiguity in the ligand binding pose. Could the authors improve the current cryo-EM map quality by further processing or collecting additional data and refining the model properly? Furthermore, why not directly assemble the complex by adding exogenous 3-OH-C12 or other FFAs for structure determination?

Response: We thank the reviewer for the advice. Indeed, the density for 3-OH-C12 in our previous model is not strong enough to definitively build either the exact conformation or the whole structure of the speculative 3-OH-C12. We cannot get better resolution with further structure refinement, possibly due to the limited quality of our previous complex sample and data collection. Following the reviewer's suggestion, we have assembled the GPR84 complex activated by adding exogenous 3-OH-C12 and obtained a map of the complex at an overall resolution of 2.89 Å (Fig. S4). Fortunately, the 3-OH-C12 in the new structure matches well with our previous model (Fig. S4). The newly-obtained 3-OH-C12 bound GPR84 complex present better density for the bound 3-OH-C12 and the surrounding residues from GPR84, which structurally provides a reliable template for understanding the recognition of 3-OH-C12 by GPR84.

Reviewer #1: 3. Similarly, the EM map cannot help the authors to fix the orientation of the pyridine ring in LY237. According to the analysis of LY237 binding, the nitrogen on the pyridine ring has a negligible effect. Interestingly, upon binding to the receptor, the pyridine ring rotates 180 degrees in some snapshots in the MD video. If a benzyl ring could replace the pyridine? Is there any possibility to further improve the potency or specificity of LY237 by strengthening the interaction in this part?

Response: We thank the reviewer for the careful inspection of our structure and manuscript. In one of our previous papers, we synthesized and evaluated a series of alkylpyrimidine-4,6-diol derivatives as novel GPR84 agonists (Ref-5), we found that the length of alkyl chain, the type

of linker group, the substitutions on the pyrimidine ring and replacement of the pyrimidine ring all influence the agonist activity of GPR84. Finally, we identified 6-nonylpyridine-2,4-diol (LY237 in this study) as a potent and selective GPR84 agonist, with an EC₅₀ of 0.189 nM. We also synthesized LY236, which has a benzyl ring instead of the pyrimidine ring in the same position as LY237 (Fig. R1a). LY236 activates GPR84 with an EC₅₀ of 0.435 μM, which is over 2,000-fold less potent than LY237 (Fig. R1b), illustrating non-negligible effect of the nitrogen of the pyridine ring in binding and activating GPR84.

Fig. R1 SAR study and docking results of LY236 and LY237.

In our LY237-bound structure, we can see that the nitrogen points to a relatively polar pocket formed by the surrounding residues, including Y69, T359, and W360, in our LY237-bound structure. This is energy-favored, which cannot be satisfied by the opposed pocket mostly formed by hydrophobic residues, including L100, F101, and F170. In addition, although the absence of strong polar interactions with the nitrogen on the pyridine ring of LY237 in our structure, we still cannot rule out the possibility of the structure water-mediated polar interactions. Unfortunately, we are not able to model additional water molecular in the LY237-bound pocket or discuss the potential polar interactions with the nitrogen on the pyridine ring of LY237, due to the limited resolution of our LY237 structure. To further identify the orientation of LY237 in GPR84 pocket, we performed molecular docking of LY237 into the binding pocket of the GPR84 structure. In the docking model, the pyridine nitrogen of LY237 face to the hydrophilic sub-pocket constituted by residues including Y69, T359, and W360 from GPR84 structure (Fig. R1c), which is consistent with our current model of LY237 in GPR84. Chemically, the two hydroxyl groups of LY237 are capable of forming stronger polar interactions with the polar pocket of GPR84, owing to the greater hydrophilic effect of the pyrimidine ring relative to the benzyl ring. This could be a contributing factor to the over 2,000-fold higher potency of LY237 compared to LY236 in activating GPR84.

References

Ref-5. Liu, Y. et al. Design and Synthesis of 2-Alkylpyrimidine-4,6-diol and 6-Alkylpyridine-2,4-diol as Potent GPR84 Agonists. ACS medicinal chemistry letters 7, 579-583, doi:10.1021/acsmchemlett.6b00025 (2016).

Reviewer #1: 4. Given the importance of the N-terminus and ECL-2 loop conformations, please provide the density map of the N-terminus and ECL-2 loop in the figure to better present the

interactions.

Response: We thank the reviewer for the advice, and we have displayed the density map of the N-terminus and ECL-2 loop in Figure. S2 in our revised manuscript to better present the interactions.

5. Some minor points: a few panels and figure legends should be adjusted. For example,

Reviewer #1: (a) The space-filling representations of disulfide bonds shown in Fig. 1c and S3 are a little bit crowded and better to change to sticks;

Response: We thank the reviewer for the advice, and we have changed the sticks radius of disulfide bonds which were shown in Fig. 1c and S3.

Reviewer #1: (b) The legends for Fig 2b-d are confused.

Response: We are sorry for the ambiguity, and we have adjusted the legends for Fig 2b-d in our revised manuscript.

Reviewer #1: (c) The line styles and colors of Fig 3f are inconsistent with the description in legends;

Response: We are sorry for the mistake, and we have adjusted the legends for Fig 3 in our revised manuscript.

Reviewer #1: (d) The residue label in Fig 5d is missing;

Response: We are sorry for the mistake, and we have added residue label in Fig 5d in our revised paper.

Point-by point response to Reviewer #2:

Reviewer #2: In this manuscript, Liu et al., have obtained and characterized the cryoEM structure of GPR84 bound to LY237 or 3-OH-C12 agonists in the presence of a Gi protein. The structure is supported by the wealth of mutagenesis data and explored by molecular dynamics simulations. From this integrated approach, the authors provide the unique features of a receptor structure, ligand binding pathway and interactions, receptor activation, and G-protein coupling. The manuscript is of a high standard, but several issues should be addressed before its publication:

Response: We sincerely appreciate the time and efforts that the reviewer has taken to provide insightful feedback that has help us strengthen our manuscript. With all the issues addressed, we think that our revised paper is much improved with better quality and resolution of our structures.

Reviewer #2: 1. The section on 3-OH-C12. The authors should elaborate more on how they obtained the structure of GPR84 bound to 3-OH-C12. The current text is not convincing that they tried to get an empty receptor but obtained the receptor bound to 3-OH-C12 by an accident not adding the ligand. How comes such a rare lipid which is not typical for a lipid continuum used could sit in the protein mixture for cryoEM?

Response: We are grateful for the kind comment, and thank the reviewer for the suggestion.

Indeed, we got our previous 3-OH-C12 structure without exogenous ligands applied. It has been reported that 3-OH-C12 is formed during β -oxidation in peroxisomes and its plasma concentration can reach micromolar order in mitochondrial fatty acid β -oxidation disorders (Ref-1 and Ref-2). We speculate the 3-OH-C12/GPR84 complex formed in our insect cell protein expression system.

In order to better understand recognition of 3-OH-C12 by GPR84, we have successfully assembled the GPR84 complex activated by adding exogenous 3-OH-C12 and obtained a map of the complex at an overall resolution of 2.89 Å. The 3-OH-C12 in the new structure matches well with our previous model (Fig. S4). The newly-obtained high-resolution 3-OH-C12 bound GPR84 complex present better density for the bound 3-OH-C12 and the surrounding residues from GPR84, which structurally provides a reliable template for modeling MCFA recognition by GPR84. To avoid any misunderstanding, we used the new 3-OH-C12 structure as a template to investigate the mechanism of GPR84 activation by medium-chain fatty acid chains in our revised paper.

Reviewer #2: 2. Line 251, ‘The closed ECL2 in the cryoEM structure spontaneously opened in 3 of 8 LiGaMD simulations’ please clarify here, the loop opens spontaneously in the absence of the ligand nearby in three simulations, and other five it opens when the ligand comes close and under influence of the ligand. I assume that the loop should be opened in all the simulations during the entrance process otherwise the ligand cannot enter.

Response: We are sorry for previous vague expression. We meant that ECL2 opened in 3 LiGaMD simulations even without the interaction with LY237, in the other 5 simulations, ECL2 did not open during 1000 ns simulations. Also, LY237 did not enter the pocket in all trajectories. As you expected, LY237 only started the entrance process in the 3 trajectories whose ECL2 is opened. As shown in Figure R1, the difference in the Z coordinate between the C α of V164^{ECL2} in simulations and cryo-EM structure (h_{ECL2}) stably became more than 2.5 Å (open threshold) in trajectories 1, 4, and 8. Thus, we stated that it opened in 3 of 8 trajectories. We have adjusted the expression in the revised manuscript as follows to prevent misunderstanding:

The closed ECL2 in the cryo-EM structure spontaneously opened in 3 of 8 LiGaMD simulations, in which h_{ECL2} stably became more than 2.5 Å (Fig. S6b). In the other 5 trajectories, ECL2 did not open.

Figure S7: The h_{ECL2} in each trajectory during 8 LiGaMD simulations. Red dashed line indicated the open threshold (2.5 Å).

Reviewer #2: 3. Please show the distances (hligand, hECL2) that were monitored in MD simulations in Figure 4.

Response: We thank the reviewer for the advice, and we have added the distances (hligand, hECL2) in Figure 4 in our revised paper

Reviewer #2: 4. MD simulations methods. Please specify the values of a boost potential to bring the ligand inside the receptor and to enhance the sampling of the receptor conformations, and how these values were selected, and convergence achieved.

Response: We thank the reviewer for the advice. The values for boost potentials in LiGaMD simulations were dependent on the value of current potential energy and the sampling process. We have adjusted the description in method to show it and summarize the values in Table R2 and R3. The convergence of such parameters was shown in Figure S11, S12.

In each process, 1.2 ns conventional MD was first applied to the system to further equilibrate the system. During the 1.2 ns simulations, potential energies were not collected for calculating their LiGaMD parameters. Then, 12 ns conventional MD was used to calculate LiGaMD acceleration parameters, including the non-bonded potential energy of the bound ligand (E_{nb}) and remaining potential energy of the rest of the system (E_{rp}). Then, the boost potential ΔV for non-bonded and potential energy was calculated by Eq. (1).

$$\Delta V = \begin{cases} \frac{1}{2}k_0(E_{th} - V)^2, & V < E_{th} \\ 0, & V > E_{th} \end{cases} \quad (1)$$

where k_0 was the harmonic constant and E_{th} was the threshold energy. For non bonded potential energy, k_0 and E_{th} were calculated by Eq. (2) and Eq. (3), respectively.

$$k_0 = \left(1 - \frac{\sigma_0}{\sigma_V}\right) \frac{V_{max} - V_{min}}{V_{avg} - V_{min}} \quad (2)$$

$$E_{th} = V_{min} + \frac{V_{max} - V_{min}}{k_0} \quad (3)$$

For remaining potential energy, k_0 and E_{th} were calculated by Eq. (4) and Eq. (5), respectively.

$$k_0 = \min\left(1.0, \frac{\sigma_0}{\sigma_V} \times \frac{V_{max} - V_{min}}{V_{max} - V_{avg}}\right) \quad (4)$$

$$E_{th} = V_{max} \quad (5)$$

where V_{max} , V_{min} , V_{avg} , and σ_V are the maximum, minimum, average and standard deviation of the corresponding potential energy value, and σ_0 was 6 kcal/mol as guided in Amber tutorial. These values were used during the following 1.2 ns LiGaMD equilibrium runs. Then, 60 ns parameter sampling were applied for updating these potential energy values under LiGaMD condition. The parameters for each run were listed in Table 1 and Table 2. During the 73.2 ns sampling runs, all the parameters have converged as Fig. S11, S12.

Table R2. Parameters for non-bonded potential energy boosting

Repeat	V_{max} (kcal/mol)	V_{min} (kcal/mol)	V_{avg} (kcal/mol)	σ_V (kcal/mol)
1	43.0303	-98.2983	-29.7762	7.9339
2	39.9781	-92.442	-38.1154	8.856
3	128.9275	-95.8721	-31.8457	7.4627
4	123.1266	-92.8427	-33.1884	7.9531
5	32.2203	-88.7339	-33.5473	8.6824
6	53.8041	-101.869	-40.5985	8.3789
7	97.4856	-95.6821	-27.5663	6.0549
8	88.0037	-90.6698	-31.9644	7.6115

Table R3. Parameters for remaining potential energy boosting

Repeat	V_{max} (kcal/mol)	V_{min} (kcal/mol)	V_{avg} (kcal/mol)	σ_V (kcal/mol)
1	-188193.0264	-195099.5019	-188535.2496	63.9921
2	-188122.5559	-195222.3564	-188501.6646	64.5597
3	-186760.9133	-195289.0602	-187295.3250	69.3229
4	-187293.8718	-195040.6451	-187751.5840	66.7513

5	-188080.277	-195056.8473	-188440.3333	64.7048
6	-187432.2234	-195201.6755	-187889.6753	67.3306
7	-187140.8344	-195235.8072	-187596.0806	67.3265
8	-186282.0971	-195124.0269	-186878.0558	68.9874

Figure S11. The time-course figure for V_{max} , V_{min} , V_{avg} , and σ_V of ligand binding potential energy during 73.2 ns parameter sampling.

Figure S12. The time-course figure for V_{\max} , V_{\min} , V_{avg} , and σ_V of remaining potential energy during 73.2 ns parameter sampling.

Reviewer #2: 5. Line 256. ‘Mutations to disrupt these interactions’ What mutations do authors mean? Mutations of V164, C166 or C11. Please specify.

Response: We are sorry for the omission, and we have specified the detailed amino acid in our revised manuscript.

Reviewer #2: 6. Lines 419-420. The disulfide bridge between C11 and C166 has been predicted by AlphaFold and used to explain the SAR of triazine antagonists. Please make references to AlphaFold and Investigating the Structure-Activity Relationship of 1,2,4-Triazine G-Protein-Coupled Receptor 84 (GPR84) Antagonists. Mahindra A, Jenkins L, Marsango S, Huggett M, Huggett M, Robinson L, Gillespie J, Rajamanickam M, Morrison A, McElroy S, Tikhonova IG, Milligan G, Jamieson AG. *J Med Chem.* 2022 Aug 25;65(16):11270-11290.

Response: We thank the reviewer for the suggestion, and we have read this paper carefully and cited it in our revised manuscript.

Reviewer #2: 7. Lines 424-426. The anchoring role of R172 for agonist binding has been

previously described and the cryoEM structure confirms its direct involvement in coordinating the agonist. Please cite two references: Tikhonova I.G. Application of GPCR structures for modelling of free fatty acid receptors. *Handb. Exper. Pharmacol.* 2017;236:57–77. Mahmud Z.A., Jenkins L., Ulven T., Labéguère F., Gosmini R., De Vos S., Hudson B.D., Tikhonova I.G., Milligan G. Three classes of ligands each bind to distinct sites on the orphan G protein-coupled receptor GPR84. *Sci. Rep.* 2017;7:17953.

Response: We thank the reviewer for the advice, we have read these two papers carefully and cited them in our revised manuscript.

Reviewer #2: 8. Line 426. ‘‘Previous reports’ please add the references.

Response: We thank the reviewer for the suggestion, and we have added the references in our revised paper.

Reviewer #2: 9. Video: please visualise the key polar residues (R349, H352 and R172) stabilizing various states during the binding process.

Response: We thank the reviewer for the advice, we have shown these key polar residues when they interact with LY237 in the revised video S1.

Reviewer #2: 10. Line 249, applied ->used.

Response: We apologize for the typo and grammatical errors in the original manuscript and we thank the reviewer for the detailed comments on issues with English, and we have gone through the whole text to correct language errors.

Reviewer #2: 11. Line 253, indicating an opening tendency -> remove.

Response: We are sorry for the typo, and we thank the reviewer for the careful inspection of our structure and manuscript.

Point-by point response to Reviewer #3:

Reviewer #3: This manuscript by Liu et al. reported 3.2 Å-resolution Cryo-EM structure of GPR84•Gi complex, which was reported to be involved in immunity by activating macrophages. While several Cryo-EM lipid-sensing GPCR structures coupled with Gi have already been reported, such as SIPR and LPA1 receptors, and activation mechanisms are similar, the outstanding progress in this manuscript is ligand entry mechanism involving lid opening revealed by MD simulation and complementary functional analysis. First, the authors solved the GPR84 complex structure without any ligand nor agonist, they found EM density in the orthosteric pocket, which resembles that of MCFAs such as 3-OH-C12, a putative endogenous ligand for GPR34. Thus they solved the activated structure of GPR84 in complexes with an endogenous ligand and super agonist, LY237. Intermediate structures of MD simulation were confirmed by functional analysis, which is very original and consistent with the simulation. The GPCR activation mechanism as well as Gi interaction mechanism are almost similar to the other class-I GPCRs. The structural analysis is solid and the paper is well written. Therefore, this reviewer recommends publication of this paper in *Nature Communications*, once the authors properly addresses the following minor comments:

Response: We really appreciate the insightful and positive comments by the reviewer. We want to thank the reviewer for his/her tremendous efforts in evaluating our manuscript and for the comments on the importance and biological significance of the ligand recognition and activation of the medium-chain fatty acid-sensing receptor GPR84 again.

Reviewer #3: 1. As the LY237 complex structure is the main story of this manuscript, the author should present structural formula of LY237 in main Fig. 1.

Response: We thank the reviewer for the advice, and we have displayed the structural formula of LY237 in Fig. 1b in our revised paper.

Reviewer #3: 2. In line 164, the authors should cite reference of LPA1 (2 papers) and S1PR.

Response: We are sorry for the omission, and we have added the references in our revised paper.

Reviewer #3: 3. In lines 272-273, there are some typo, since the following residue numbering is inconsistent with Fig. 4 ?; V165 should be V164 and A356 should be A365.

Response: We are sorry for the omission, and we have corrected the amino acid numbers in our revised paper.

REVIEWERS' COMMENTS

Reviewer #1 (Remarks to the Author):

I appreciate the authors for improving the manuscript based on my suggestions. I'm satisfied with the current version and recommend for publication.

Reviewer #2 (Remarks to the Author):

The authors have addressed all my comments. I have nothing to add. I recommend publishing it.

Manuscript ID: Nature Communications Manuscript (NCOMMS-22-49620A)

Title: Structural insights into ligand recognition and activation of the medium-chain fatty acid-sensing receptor GPR84

We want to thank both reviewers for their tremendous efforts in evaluating our manuscript again. In the following sections, we provide point-by-point responses to the comments by both reviewers of our revised paper. The reviewer's comments are in **black** and our responses are in **blue**.

Point-by point response to Reviewer #1:

Reviewer #1 (Remarks to the Author): I appreciate the authors for improving the manuscript based on my suggestions. I'm satisfied with the current version and recommend for publication.

Response: We are happy to hear that all comments have been addressed. We also greatly appreciate the Reviewer for all the constructive comments and suggestions.

Point-by point response to Reviewer #2:

Reviewer #2 (Remarks to the Author): The authors have addressed all my comments. I have nothing to add. I recommend publishing it.

Response: We are glad to hear that all comments have been addressed and highly appreciate the Reviewer for all valuable comments.